# Para-infectious brain injury in COVID-19 persists at follow-up despite attenuated cytokine and autoantibody responses

To understand neurological complications of COVID-19 better both acutely and for recovery, we measured markers of brain injury, inflammatory mediators, and autoantibodies in 203 hospitalised participants; 111 with acute sera (1–11 days post-admission) and 92 convalescent sera (56 with COVID-19-associated neurological diagnoses). Here we show that compared to 60 uninfected controls, tTau, GFAP, NfL, and UCH-L1 are increased with COVID-19 infection at acute timepoints and NfL and GFAP are significantly higher in participants with neurological complications. Inflammatory mediators (IL-6, IL-12p40, HGF, M-CSF, CCL2, and IL-1RA) are associated with both altered consciousness and markers of brain injury. Autoantibodies are more common in COVID-19 than controls and some (including against MYL7, UCH-L1, and GRIN3B) are more frequent with altered consciousness. Additionally, convalescent participants with neurological complications show elevated GFAP and NfL, unrelated to attenuated systemic inflammatory mediators and to autoantibody responses. Overall, neurological complications of COVID-19 are associated with evidence of neuroglial injury in both acute and late disease and these correlate with dysregulated innate and adaptive immune responses acutely.

At the beginning of the COVID-19 pandemic, neurological complications occurred in a significant proportion of hospitalised patients[1] and even in those with mild COVID-19 infection[2]. While these neurological 'complications' were often mild (headache and myalgia), it became clear that more significant neurological sequelae were observed, including encephalitis/encephalopathies, Guillain Barre Syndrome, seizure, and stroke[3–6].

Although in vitro studies show that SARS-CoV-2 can infect neurons and astrocytes[7,8], autopsy studies indicate that direct viral invasion is unlikely to be a cause of neurological dysfunction in vivo[9]. Post-mortem studies failed to detect viral infection of the brain by immunohistochemistry in the majority of cases, and viral qPCR levels were often low and may simply have reflected viraemia[10–12]. In addition, virus and/or anti-viral antibodies were rarely found in cerebrospinal fluid (CSF)[13]. Thus, it seems more likely that the virus affects the brain indirectly. This could be through peripherally generated inflammatory mediators, immune cells, autoantibodies and/or blood brain barrier changes associated with endothelial damage[14,15]. Immune infiltrates have been found in autopsy studies, including neutrophils and T cells, although agonal effects could not be excluded[16]. On the other hand, elevated IL-6 levels in sera and CSF have been associated with neurological complications, including meningitis, thrombosis, stroke, cognitive and memory deficits, regardless of respiratory disease severity[17–20]. One study found that the brain injury markers NfL and GFAP, and inflammatory cytokines were elevated in COVID-19 and scaled with severity[21–25]; another study showed that baseline CSF NfL levels correlated with neurological outcomes at follow-up[26] but overall, the relationships between these immune mediators and markers of brain injury and neuropathology remains to be fully explored. Finally, specific neuronal autoantibodies have been reported in some neurological patients raising the possibility of para- or post-infectious autoimmunity[14,27].

e-mail: benmic@liverpool.ac.uk

To assess the relationship between host immune response and markers of brain injury with neurological injury, we studied two large, multisite cohorts which, in combination, provided acute, early and late convalescent sera from COVID-19-positive (COVID+ve) participants. We measured brain injury markers, a range of cytokines and associated inflammatory mediators, and autoantibodies in these samples, and related them to reduced levels of consciousness (defined as a Glasgow Coma Scale Score [GCS] GCS ≤ 14) in the acute phase, or the history of a neurological complication of COVID-19 in convalescent participants. We tested the hypothesis that immune mediators would correlate with brain injury markers and reveal a signature of neurological complications associated with COVID-19.

## Results

### COVID-19 results in acute elevation of serum markers of brain injury, more so in participants with abnormal Glasgow coma scale (GCS) score

We used sera from the International Severe Acute Respiratory and emerging Infection Consortium Clinical Characterisation Protocol United Kingdom (ISARIC CCP-UK) study, obtained 1–11 days post admission, that included 111 participants with COVID-19 of varying severity and 60 uninfected healthy controls (labelled Control). Participants were stratified by normal ($n = 76$) or abnormal ($n = 35$) Glasgow Coma Scale scores (labelled GCS = 15 or GCS ≤ 14, respectively) to provide a proxy for neurological dysfunction (Fig. 1a). GFAP (glial fibrillary acidic protein, marker of astrocyte injury), UCH-L1 (a marker of neuronal cell body injury), and NfL (neurofilament light) and Tau (both markers of axonal and dendritic injury) were measured. Overall, serum levels of NfL, GFAP, and total-Tau (tTau) were significantly higher in COVID-19 participants compared to the uninfected healthy controls but, as shown in Fig. 1b–e, those participants with abnormal GCS scores had higher levels of NfL and UCH-L1 than those with normal GCS scores. Thus, all four markers of brain injury were raised in COVID-19 participants (both GCS = 15 and GCS ≤ 14) but, in addition, axonal and neuronal body injury biomarkers discriminated between participants with and without reduced GCS.

### Markers of brain injury remain elevated in the early and late convalescent phases in participants who have had a CNS complication of COVID-19

To ask whether these findings persisted in participants recovering from COVID-19-related neurological complications, ninety-two COVID-19 participants were recruited to the COVID-Clinical Neuroscience Study (COVID-CNS), 56 who had had a new neurological diagnosis that developed as an acute complication of COVID-19 (group labelled "neuro-COVID"), and 36 with no such neurological complication (group labelled "COVID", Fig. 1f, Table 1, Supplementary Tables 1 and 2). When compared to the same healthy controls ($n = 60$), across all timepoints, both COVID-19 subgroups (COVID and neuro-COVID) showed increased levels of NfL, GFAP, and tTau (but not UCH-L1 (Fig. 1g–j, Supplementary Table 1)). Furthermore, participants recovering from neuro-COVID had significantly higher levels of NfL, and a trend towards higher levels of tTau, than the COVID participants (Fig. 1g, j). Highest NfL serum levels were present in participants with cerebrovascular conditions, whereas tTau was elevated in participants with cerebrovascular, CNS inflammation and peripheral nerve complications (Fig. 1k, l). NfL remained significantly elevated in a multiple regression model adjusted for age (Supplementary Fig. 1a, b). We then separately compared the two cohorts at early and late convalescent follow-up periods (less than and over six weeks after admission respectively). NfL and GFAP levels remained elevated in all COVID-19 participants in the convalescent period, but only remained elevated beyond 6 weeks in participants who had suffered an acute neurological complication (neuro-COVID, Fig. 1m–p; Supplementary Fig. 1c). The presence of elevated brain injury markers in the acute phase of COVID-19 confirms previous findings[14], but the elevated levels of NfL and GFAP in those who are convalescent from acute neurological complications suggest ongoing neuroglial injury.

### Clinical and brain injury markers evidence of neurological insult levels are associated with levels of innate inflammatory mediators in the acute phase of COVID-19

To explore whether the acute and persistent elevation of markers of brain injury observed in participants with COVID-19 was associated with an acute inflammatory response, we measured a panel of 48 inflammatory mediators in serum at the same timepoints. In the ISARIC samples, six mediators were significantly higher in participants with an abnormal GCS than in those with a normal GCS (interleukin [IL]-6, hepatocyte growth factor [HGF], IL-12p40, IL-1RA, CCL2 and macrophage colony stimulating factor [M-CSF]), indicating increased innate inflammation (Fig. 2a, Supplementary Fig. 2a). Pearson's correlation tests identified correlations between these significant immune mediators in an interrelated pro-inflammatory network (Fig. 2b, c), and unsupervised Euclidean hierarchical cluster analysis revealed clusters of pro-inflammatory mediators elevated together (Fig. 2d). The first cluster incorporated the IL-1 family (including IL-1RA), interferons and M-CSF, and the second cluster included IL-6, CCL2, CXCL9, HGF, and IL-12p40 (boxes in Fig. 2d). Brain injury biomarkers correlated with elevations in these inflammatory mediators: GFAP and UCL-H1 correlated with a number of mediators in the first cluster, whereas tTau and NfL correlated strongly with HGF and IL-12p40 in the second cluster (Supplementary Table 3).

A more stringent analysis of median-centred cytokine data (which corrected for between participant skewing of mediator levels) confirmed that HGF and IL-12p40 were higher in the abnormal GCS COVID-19 participants, and correlated with cognate NfL levels (Supplementary Table 4). Taken together these data suggest that activation of the innate immune system was related to both clinical and blood marker evidence of CNS insult.

### Inflammatory mediators are not elevated across the participant cohort at late timepoints after COVID-19; but late tTau elevations correlate with levels of several inflammatory mediators

In contrast to the acute data, the levels of cytokines and associated mediators were lower when measured during the convalescent periods even in those who had suffered neurological complications of COVID-19 (group labelled "neuro-COVID". Supplementary Fig. 2b). The correlations between cytokines and associated mediators no longer displayed the same tight clusters (Fig. 2e, f). GFAP remained elevated during the convalescent phase of neurological complications (Fig. 1p) but did not show correlations with the inflammatory mediators. Similarly NfL was higher overall in those with neurological complications (Fig. 1n) but there were no significant correlations with inflammatory mediators (Fig. 2f). However, tTau remained elevated overall in those with neurological complications ((1.7 (1.3, 2.2) pg/mL versus 1.3 (1.1, 1.9) pg/mL)) and levels correlated with eight immune mediators including CCL2, IL-1RA, IL-2Rα and M-CSF along with CCL7, stem cell factor (SCF), IL-16 and IL-18 (Fig. 2f, Supplementary Table 5, Supplementary Fig. 2c). This last association was specific to the late phase of the illness and was not found in acute COVID-19.

### Cytokine networks are significantly altered in participants with neurological complications of COVID-19: both acute encephalopathy, and those recovering from a neurological complication

We used graph theoretical approaches to compare these cytokine networks between participants with: acute COVID-19 and normal GCS; acute COVID-19 with altered consciousness (GCS ≤ 14), and convalescent participants recovering from a neurological complication of COVID-19 (neuro-COVID). Participants with both neurological

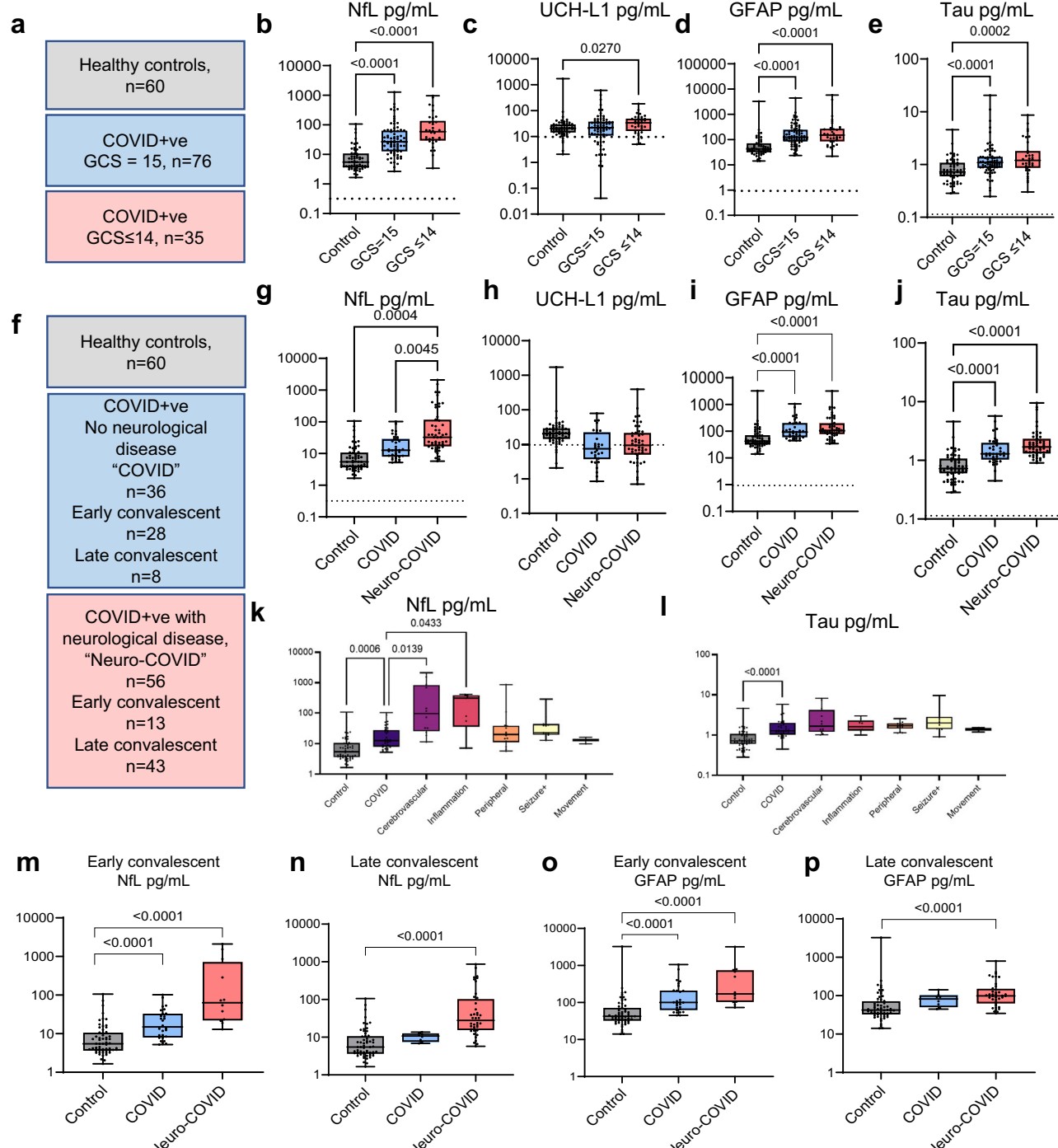

**Fig. 1 | Brain injury markers are elevated acutely in COVID-19 participants with an abnormal Glasgow coma scale score (GCS) and in participants who experienced a neurological complication associated with COVID-19. a** The acute ISARIC cohort included Day 1–11 hospital admission timepoints. **b–e** Acute serum brain injury markers were assessed by Simoa: **b** NfL, **c** UCH-L1, **d** GFAP, and **e** tTau. All four were elevated in COVID-19 cases with normal Glasgow coma scale scores (GCS) relative to controls overall. Dotted lines show lower limit of quantification (LLOQ). **f–j** The Simoa analyses were performed for the sera from the COVID-CNS COVID and neuro-COVID groups at early and late convalescent timepoints (**g**, **j**) showing persistence of NfL, GFAP, and tTau in COVID participants, with NfL higher in neuro-COVID than COVID participants. **k** Within the combined early and late convalescent COVID-19 neurological cases, the highest levels of NfL were observed in participants who had suffered a cerebrovascular event at the time of SARS-CoV-2 infection. **l** tTau levels were raised in the cerebrovascular, CNS inflammatory and seizure conditions. **m**, **n** Serum NfL remained elevated in both the early (<6 weeks from positive SARS-CoV-2 test) and late convalescent phases (>6 weeks) in neuro-COVID compared to COVID non-neurological cases. **o**, **p** GFAP was elevated in neurological cases in the early and late convalescent phase. Box and whisker plots show all data points with median as centre line with 25th and 75th percentiles. Sample sizes shown in (**a**) and (**f**). Group comparisons are by Kruskal–Wallis test with Dunn's post-hoc multiple comparison test, no statistical comparison made for panel (**h**) as medians were at LLOQ.

**Table 1 | Clinical characteristics of healthy controls and COVID-CNS participants**

| Clinical characteristics | | Control (n = 60) | COVID (n = 36) | Neuro-COVID (n = 56)[a] |
|---|---|---|---|---|
| Age | Mean (SD) | 48 (18) | 51 (17) | 58 (13) |
| | Median (Q1,Q3) | 50 (32,62) | 52 (34,65) | 61 (48,67) |
| Gender | Male n (%) | 21 (35%) | 21 (58.3%) | 36 (64.3%) |
| Sampling time (days)[b,c] | Median (Q1,Q3) | | 8 (4,14) | 148 (52,272) |
| COVID severity[c,d] | Median (Q1,Q3) | | 5 (5,5) | 7 (5,8) |

[a]Neuro-COVID group comprises cases of: cerebrovascular conditions (21%), CNS inflammation (16%), movement disorders (5.4%), seizures and other CNS conditions (30%), and peripheral nervous system conditions (27%).
[b]Sampling time in days between the first COVID+ve test and serum sample.
[c]Two Neuro-COVID participants with approximate COVID-infection timing and nine with indeterminate severity.
[d]As per COVID-19 WHO severity score from 0 to 10.

consequences of COVID-19 (GCS ≤ 14) and Neuro-COVID both showed cytokine networks that were different from COVID-19 participants with no neurological problems (Fig. 2b, c, e; $p < 0.001$, Steiger test), suggesting a specific dysregulated innate immune response that is associated with neurological complications of COVID-19. Further pathway analyses using the KEGG enrichment scores on the significantly different cytokines, revealed many commonalities with other inflammatory syndromes (Supplementary Fig. 3a, b). Interestingly, cytokine profiles of the neurological complications groups from both the ISARIC and COVID-CNS cohort led to JAK-STAT signalling being a significant involved pathway which would be amenable to immunomodulation, for example, by tofacitinib, which has been shown to reduce mortality in COVID-19[28].

### COVID-19 is associated with an acute adaptive immune response overall, which includes antibodies to viral antigen and CNS autoantigens in those with abnormal GCS scores

Given past reports of autoantibody responses following COVID-19[14,27], we sought evidence of similar dysregulated adaptive immune responses in our participant cohorts. We used a bespoke protein microarray of 153 viral and tissue proteins to measure IgM (Fig. 3a–d) and IgG (Fig. 4a–d) reactivity in the acute phase ISARIC sera. The median fluorescence intensities for each putative antigen were normalized for each participant and the Z-scores were compared to healthy control data, to determine positive reactivity to the different antigens (with a threshold for detection set at three standard deviations above controls for each antigen; see Supplementary Table 6, Supplementary Fig. 4a). IgM and IgG responses in COVID-19 participants showed greater reactivity overall (both GCS = 15 and GCS ≤ 14), compared to the controls, with no difference in normalised fluorescence Z scores or the number of participants with IgG 'hits' (a Z-score >3) between those with normal or abnormal GCS score (Fig. 3a, b, Fig. 4a, b). However, several IgM and IgG autoantibodies, including those against the CNS antigens UCH-L1, GRIN3B and DRD2, along with the cardiac antigen, myosin light chain (MYL)-7, were present in a greater proportion of participants with an abnormal GCS score, as were antibodies to spike protein (Figs. 3c, 4c). None of the antibodies correlated significantly with levels of brain injury markers (Supplementary Figs. 4b, c, 5b, c), but they did show correlations with each other (Figs. 3d, 4d, h), suggesting a non-specific antibody response in some individuals during the acute phase.

Normalized fluorescence Z scores of serum IgM and IgG autoantibodies in the early and late convalescent samples were similar to those in the acute samples (Figs. 3e, 4e), and the IgM and IgG 'hits' were more frequent than in controls (highest in the neuro-COVID group, Figs. 3f, 4f, Supplementary Fig. 5a). However, specific autoantibody responses to MYL7, gonadotrophin releasing hormone receptor (GNRHR) and several HLA antigens were common in the neuro-COVID participants (Figs. 3g, 4g, Supplementary Fig. 5a). When the IgM and IgG hits were stratified by condition,

cerebrovascular and inflammatory conditions showed the highest number (Supplementary Fig. 5d, e). As in the acute phase, autoantibody responses did not show significant associations with brain injury markers, but did tend to correlate with each other (Fig. 4h, Supplementary Fig. 5b, c).

Finally, to explore binding to native neuronal antigens, sera from acute COVID-19 participants with CNS antigen reactivity were incubated with sections of rat brain, neurons and antigen-expressing cells. Binding to rat brain sections identified 42/185 (23%) of participants with strongly positive immunohistochemical staining (e.g. Fig. 4i) and overall, sera from the COVID+ve ISARIC participants showed more frequent binding to brainstem regions than control sera, but this did not relate to the GCS or neurological disease of the participants (Fig. 4j, Supplementary Fig. 6). In addition, from 34 selected samples tested via cell-based assays to examine for the presence of specific autoantibodies (LGI1, CASPR2, NMDAR, GABA$_B$ receptor), only one bound to the extracellular domain of the GABA$_B$ receptor (from the ISARIC cohort, Supplementary Fig. 7a, b), as expected of a pathogenic autoantibody.

## Discussion

We used several approaches to study neurological complications of COVID-19 infection. These included assessment of immune mediators and markers of brain injury in participants with and without neurological complications, both in the acute and convalescent phases after COVID-19 infection. We demonstrated increased levels of brain injury markers following COVID-19, which showed specific patterns with disease phase (acute or convalescent), and varied with the presence or absence of neurological injury or dysfunction. In the acute phase, all four brain injury markers (GFAP, NfL, tTau and UCH-L1) were elevated in participants when compared to controls, and specific markers of dendritic and axonal injury (tTau and NfL) were significantly higher in participants who showed a reduced level of consciousness (GCS ≤ 14). In the early convalescent phase (<6 weeks post-infection), GFAP, NfL, and tTau were elevated in participants recovering from COVID-19, with no differences between those who had or had not sustained a neurological complication of disease. However, at late timepoints (>6 weeks) elevations of NfL and GFAP were only seen in participants who had sustained a neurological complication of COVID-19 in the acute phase of their illness. These data suggest that clinical neurological dysfunction in COVID-19 is reflected by increases in markers of neuroglial injury, both in the acute phase and at follow-up, which are related to a dysregulated immune response, more robustly in the acute phase of illness.

In the acute phase, when compared to controls, we also observed increases in a range of inflammatory mediators (IL-6, HGF, IL-12p40, IL-1RA, CCL2, and M-CSF) in the overall cohort of COVID-19 participants, with HGF and IL-12p40 showing robust differentiation between participants with and without alterations in consciousness. By contrast, participants at the late phase after COVID-19 showed no group level elevation of inflammatory mediators. However, late elevations in tTau correlated with levels of CCL2, CCL7, IL-1RA, IL-2Rα, M-CSF, SCF, IL-16,

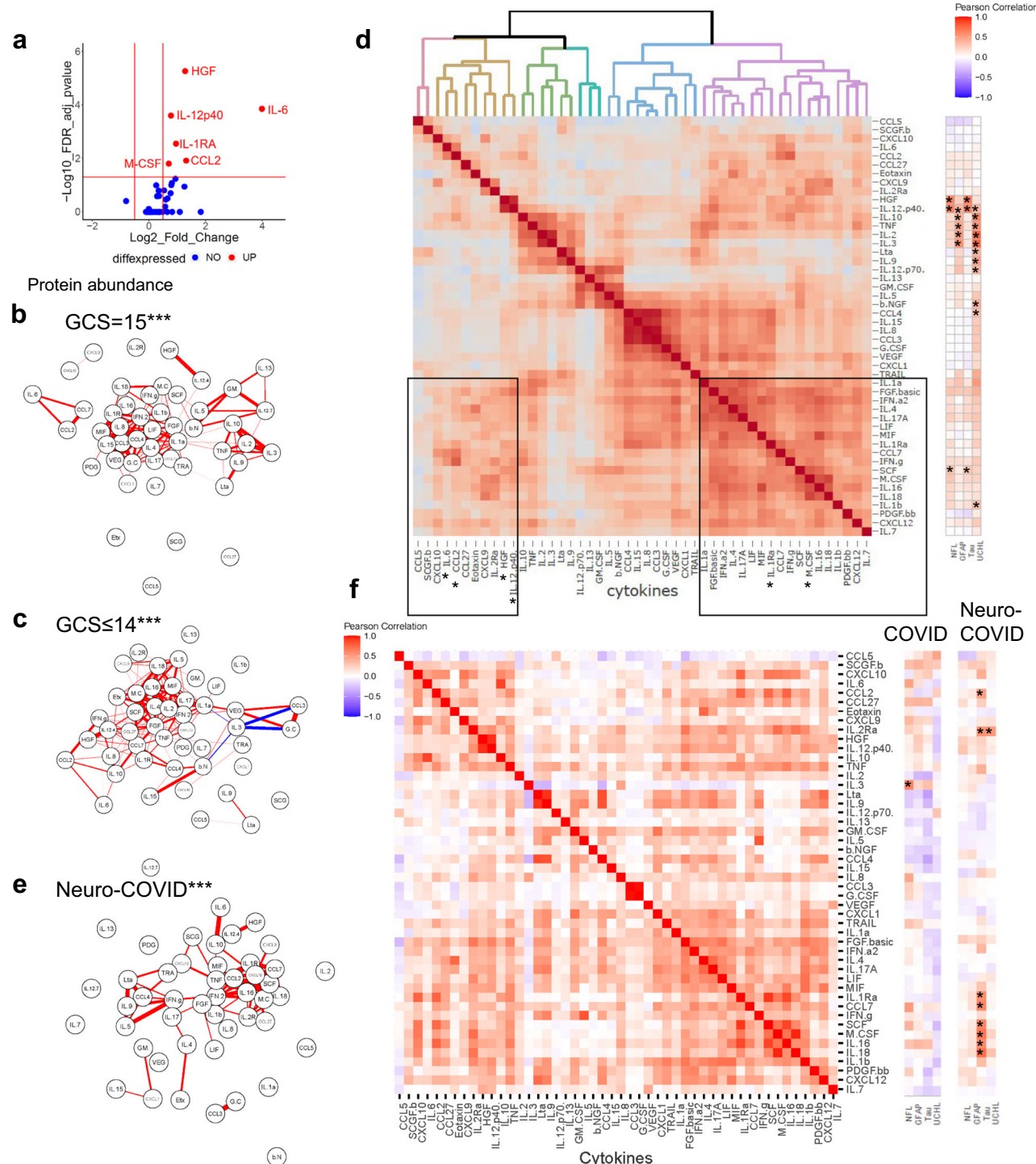

**Fig. 2 | Immune mediators are elevated acutely and correlate with different brain injury markers at different timepoints.** Serum mediators from the ISARIC and COVID-CNS cohorts were assessed by Luminex. **a** A volcano plot was generated to identify mediators which were elevated in participants with an abnormal GCS (GCS ≤ 14) compared to normal GCS (GCS = 15) and **b**, **c** a network analysis identified the highest correlations between the mediators (***significantly different from ISARIC GCS = 15 by Steiger test *p* < 0.001). **d** Unbiased Euclidean hierarchical cluster and correlation analyses identified two clusters of up-regulation of several pro-inflammatory mediators in concert. The first group included interleukin (IL)-6, IL-12p40, CCL2, CXCL9 and hepatocyte growth factor (HGF) and the second group included the IL-1 family, interferons, and macrophage colony stimulating factor (M-CSF); to the right is shown the correlations between each cytokine with the four brain injury biomarkers (significance indicated by asterisks). **e** Network analysis and heatmaps of correlations between mediators did not demonstrate the tight inter-connectedness that had been identified in acute samples and there were differences between neuro-COVID (**e**) and ISARIC GCS = 15 and GCS ≤ 14 by Steiger test (***p* < 0.001). **f** At this later stage several mediators correlated with tTau. Volcano plot used multiple two-tailed Mann–Whitney U tests with a false discovery rate set to 5%. Correlations are Pearson's coefficients (**p* < 0.05, ***p* < 0.01, ****p* < 0.001, *****p* < 0.0001).

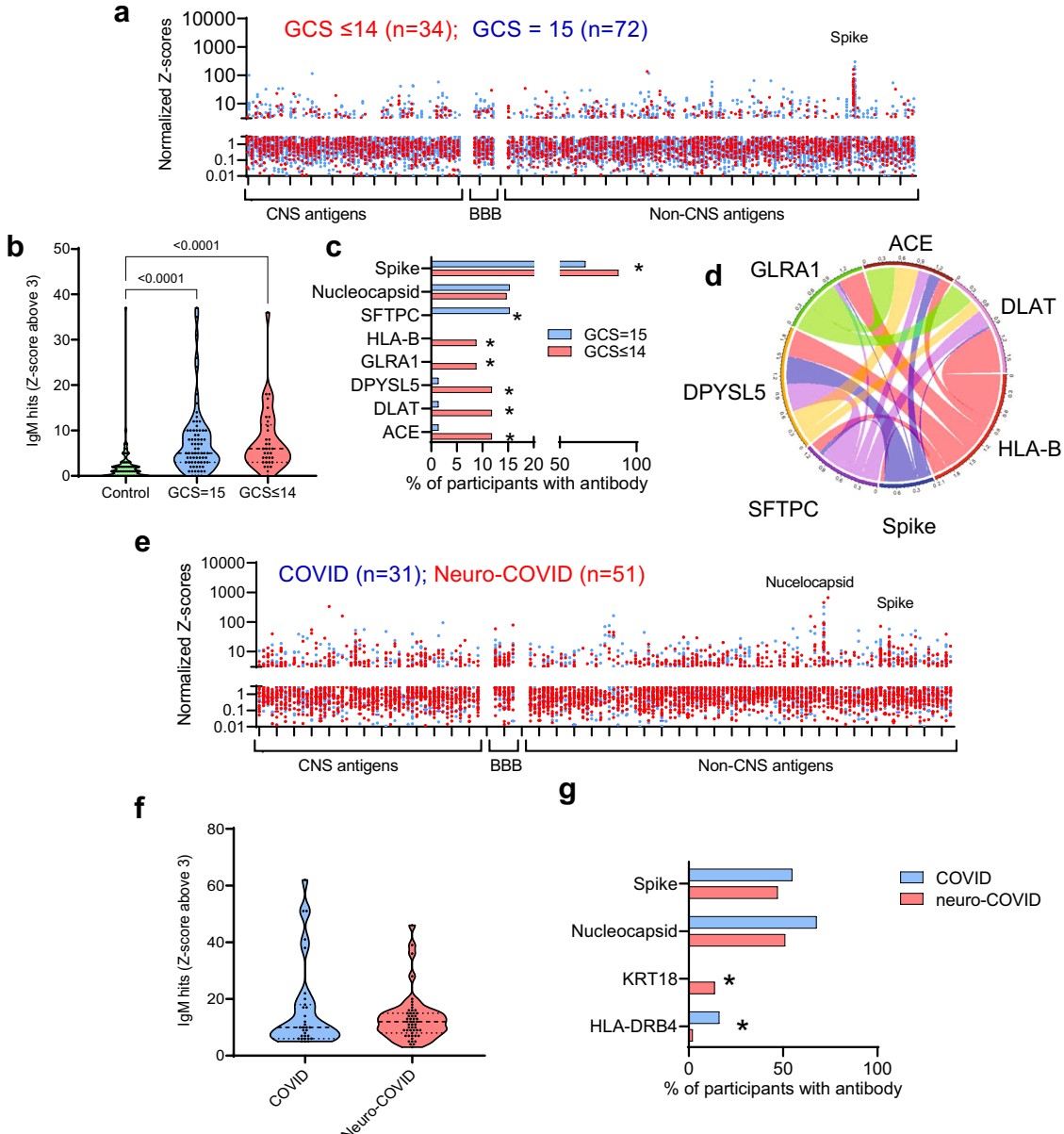

**Fig. 3 | There is an IgM antibody response in participants with COVID-19 directed at SARS-CoV-2 spike protein and against several self-antigens. a** Acute samples were tested for IgM antibodies by protein microarray with normalized fluorescence Z-scores shown. **b** COVID-19 participants showed considerably more binding 'hits' than healthy controls (fluorescence with a Z-score of 3 or above compared to controls), although overall there was no difference in the acute samples between participants with normal (GCS = 15) or abnormal GCS (GCS ≤ 14). Nevertheless, **c** COVID-19 participants with abnormal GCS (GCS ≤ 14) more frequently had raised IgM antibodies than COVID-19 participants with a normal GCS (GCS = 15), including those directed at SARS-CoV-2 spike protein (Fisher's exact tests *p < 0.05). **d** A chord diagram shows the associations between antibodies,

including those against Spike. **e** IgM antibodies were also analysed in the convalescent participants. **f** A largr proportion of COVID and Neuro-COVID participants had positive antibody 'hits' for IgM (defined by Z-score 3 and above compared to controls). **g** Of those antibodies against self-antigens identified, they were only two with different frequencies between the groups (Fisher's exact tests *p < 0.05). At this timepoint there was no significant difference in the proportion of individuals with IgM against SARS-CoV-2 spike or nucleocapsid epitopes. Violin plots show all data points with median at centre line and 25th and 75th quartile lines. Group comparisons are by Kruskal–Wallis test with post-hoc Dunn's multiple comparison test, pairwise comparisons by two-tailed Mann–Whitney U test, and correlations are Pearson's coefficients.

and IL-18, suggesting that these markers of the late innate host response were associated with persisting markers of dendritic/axonal injury markers. A network analysis showed that the repertoire of cytokine responses was different in participants both with acute reductions in GCS, or those recovering from a neurological complication of COVID-19 when compared to the GCS = 15 group.

Participants with acute COVID-19 also developed IgG autoantibody responses to a larger number of both neural and non-neural antigens, than seen in controls. These increased IgG responses

persisted into the late phase but to different antigens. While the diversity of autoantibody response did not differ between participants with and without neurological dysfunction, autoantibody responses to specific antigens, including the neural antigens UCH-L1, GRIN3B, and DRD2, were more common in participants with abnormal GCS at presentation. In the late phase, participants who had or had not experienced a neurological complication of COVID-19 were distinguished by the presence of autoantibodies to HLA antigens rather than neural antigens.

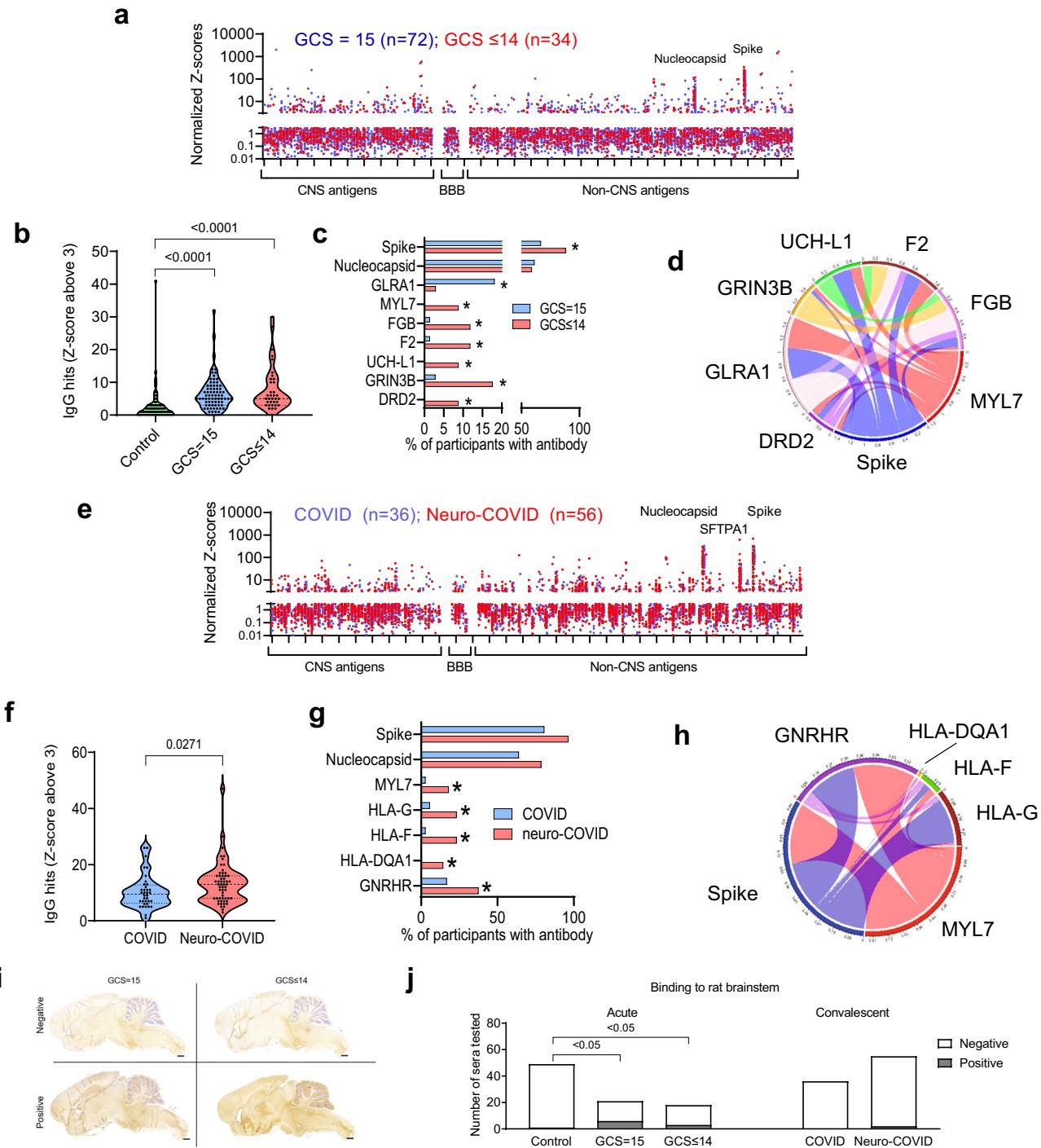

**Fig. 4 | There is an IgG antibody response in participants with COVID-19 directed at SARS-CoV-2 nucleocapsid and spike proteins and against several self-antigens. a** Acute samples were tested for IgG antibodies by protein micro-array with normalized fluorescence Z-scores shown. **b** COVID-19 participants showed considerably more binding 'hits' than healthy controls (fluorescence with a Z-score of 3 or above compared to controls) but, overall, there was no difference in the acute samples between participants with normal (GCS = 15) or abnormal GCS (GCS ≤ 14). **c** COVID-19 participants with abnormal GCS more frequently had several raised IgG antibodies than COVID-19 participants with a normal GCS, including those directed at SARS-CoV-2 spike protein and several CNS proteins ((DRD2, GRIN3B, and UCH-L1) Fisher's exact tests * p < 0.05)). **d** A chord diagram shows the association of antibodies with differences in frequency, including those against Spike. **e** IgG antibodies in early and late convalescent sera were also analysed. **f** A larger proportion of Neuro-COVID participants had positive antibody 'hits' for IgG

(defined by Z-score 3 and above compared to controls). **g** Of those antibodies against self-antigens identified, only five showed a difference in frequency between groups (Fisher's exact tests * p < 0.05). At this timepoint there was no significant difference in the proportion of individuals with IgG against SARS-CoV-2 spike or nucleocapsid epitopes. **h** A chord diagram shows the association of antibodies with differences in frequency plus anti-Spike antibody. **i** Representative images of rat brains incubated with participants' sera and screened for IHC binding of anti-human IgG to detect CNS reactivity. Scale bars = 1 mm. **j** Percentage of participant serum IgG reactivity to rat brainstems, detected by Fisher's exact test with Benjamini and Hochberg correction. Violin plots show all data points with median at centre line and 25th and 75th quartile lines. Group comparisons are by Kruskal–Wallis test with post-hoc Dunn's multiple comparison test, pairwise comparisons by two-tailed Mann–Whitney U test, and correlations are Pearson's coefficients.

These data from clinical disease provide important insights regarding the pathophysiology and pathogenesis of neurological injury, dysfunction, and disease in COVID-19. The clinical characteristics of our participant cohorts, and the elevation in brain injury markers, provide evidence of both acute and ongoing neurological injury[29]. Furthermore, the literature data on the rarity of direct CNS infection by the virus, suggest that the innate and adaptive host responses that we document should be explored as pathogenic mechanisms. The incidence of neurological cases has decreased since the first wave of the pandemic, possibly due to the use of immunosuppressants, such as dexamethasone, although this may also reflect vaccines attenuating disease and changes in the prevalence of different strains of SARS-CoV-2[30].

The inflammatory mediators that we found to be elevated in the acute phase are broadly concordant with many other publications that have examined innate immune responses in COVID-19[21,22] but there are limited data addressing associations between such responses and the development of neurological complications. It is possible that some of the risk of developing such complications is simply related to the severity of systemic infection and the host response, and it would be surprising if these were not strong contributors. However, our data suggest that acute neurological dysfunction in COVID-19 is also associated with a different repertoire of cytokine responses, with HGF and IL-12p40 showing the statistically most robust discrimination between participants with and without an abnormal GCS. HGF has important roles in brain development and synaptic biology[31] and its elevation may represent a protective/reparative response in participants with neurological injury. IL-12p40 has a core role in orchestrating Th1 responses, and has been reported to be central in the development of central and peripheral neuroinflammation, with p40 monomer subunits perhaps acting as inhibitors of the process[32–34]. Interestingly, the cytokine network that was activated in the late convalescent phase was different, potentially indicating differential drivers of neurological injury throughout the disease course. Though group level comparisons with controls showed some commonalities in inflammatory mediator increase, most notably in IL-1RA, CCL2, and M-CSF, there were many differences. The late tTau elevation that we demonstrated was significantly associated with elevations in these three mediators, but also CCL7, IL-2Rα, SCF, IL-16, and IL-18. These are all important pro-inflammatory mediators, and their association with tTau levels may reflect the persistence of a systemic inflammatory response that can enhance neuroinflammation[32,34,35].

We found a general increase in antibody production following COVID-19 infection and only a few autoantibody frequencies were different when compared by GCS or COVID versus neuro-COVID cases. Of note, absolute levels of autoantibodies were low in comparison to anti-viral antibodies that developed over the course of the acute illness, with the exception of SFTPA1. Antibodies to SFTPA1, a lung surfactant protein, have been found to correlate with COVID-19 severity[14], but these antibodies were present in only a few acute cases. HLA antibodies, on the other hand, were more frequent in Neuro-COVID than COVID participants and this requires further investigation. The autoantibodies detected in COVID-19, as in other infections, could be through molecular mimicry or bystander effects[36–39], but the lack of association of autoantibody levels with markers of brain injury is evidence against a causal role for these adaptive immune responses. Further analysis by screening the antibodies against brain antigens ex vivo revealed sporadic reactivity in both cases and controls with only the brainstem showing increased reactivity in acute COVID+ve participants; the frequencies were lower in COVID and neuro-COVID cases with no difference between them.

Our studies have several limitations including: limited clinical information on the acute participants and lack of longitudinal blood samples; in addition, the low GCS could indicate sedation for intubation, rather than CNS disease, in the acute cohort. Although we did not have COVID-19 severity scores, we did know whether participants had required oxygen or not; when data were analysed within the cohorts comparing participants who had or had not required oxygen, 5 out of 6 cytokines remained significantly elevated in the abnormal GCS group. In the COVID-CNS study where we did have in-depth clinical information, we were limited by not having acute blood samples. Nevertheless, several cytokines showed significant positive correlations with the brain injury marker tTau, and interestingly, three of them were cytokines that were significantly associated with abnormal GCS in the acute cohort (IL-1RA, CCL2, and M-CSF) highlighting a network of co-upregulated immune mediators associated with neurological complications. The commonalities in innate immune response in participants who suffered neurological dysfunction/complications, both in the acute phase and at convalescence, is underlined by the results of network analysis. Pro-inflammatory cytokines are expected to be increased in the anti-viral response, but we found that they not only correlate with COVID-19 severity, but with GCS, as well. Strengths of our study include the large cohort of participants studied with well-characterized neurological syndromes and a known range of timings since COVID-19 infection. We studied aspects of the innate and adaptive immune response as well as brain injury markers in order to discover useful markers of neurological complications over time.

Several hypotheses for how SARS-CoV-2 causes neuropathology have been tested. A prospective study of hospitalised patients showing IL-6 and D-dimer as risk factors for neurological complications implicates the innate immune response and coagulation pathways[19]. The complement pathway and microthrombosis have been associated with brain endothelial damage from the infection, and this phenotype persists months after COVID-19[40,41]. Animal models have provided key insights into COVID-19 neuropathology that warrant discussion. There have been at least two reports of viral encephalitis and neuron degeneration and apoptosis observed in non-human primates[42,43]. It is important to note that in these studies the virus was present at low amounts in the brain and predominantly in the vasculature as visualized by co-localization with Von Willebrand Factor[43]. Similar to the clinical scenario, there was no correlation of neuropathology with respiratory disease severity[43]. A recent mouse study is particularly relevant to our work and involved assessment of a mouse model that lacked direct viral neural invasion by infecting mice that were intratracheally transfected with human ACE2. This study reported increased CXCL11 (eotaxin) in mouse serum and CSF that correlated with demyelination and was recapitulated by giving CXCL11 intraperitoneally[44]; this was linked to clinical studies that showed elevated CXCL11 in patients with brain fog[44]. A combined analysis of hamster and clinical studies showed that COVID-19 led to IL-1β and IL-6 expression within the hippocampus and medulla oblongata and decreased neurogenesis in the hippocampal dentate gyrus which may relate to learning and memory deficits[45]. This was also borne out during in vitro studies that showed that serum from COVID patients with delirium lead to decreased proliferation and increased apoptosis of a human hippocampal progenitor cell line mediated by elevated IL-6[46].

In conclusion, we show evidence of quantifiable neuroglial injury markers in blood from COVID-19 infection, which is more prominent in patients with neurological dysfunction in the acute phase of the illness, and persists in the convalescent phase in patients who suffered defined acute neurological complications. These brain injury markers are associated with dysregulated systemic innate and adaptive immune responses in the acute phase of the disease, and suggest that these may represent targets for therapy.

## Methods

### Human participant studies/healthy controls and ethics information

The ISARIC WHO Clinical Characterization Protocol for Severe Emerging Infections in the UK (CCP-UK) was a prospective cohort study of

hospitalised patients with COVID-19, which recruited across England, Wales, and Scotland (National Institute for Health Research Clinical Research Network Central Portfolio Management System ID: 14152). Participants were recruited prospectively during their hospitalisation with COVID-19 between February 2020 and May 2021. The protocol, revision history, case report form, patient information leaflets, consent forms and details of the Independent Data and Material Access Committee are available online[47]. Ethical approval for CCP-UK was given by the South Central - Oxford C Research Ethics Committee in England (Ref 13/SC/0149) and the Scotland A Research Ethics Committee (Ref 20/SS/0028). We examined 111 participants with anonymized clinical data including Glasgow coma scale score and consented serum sample. ISARIC samples were collected during the acute phase (1–11 days from hospital admission). Healthy control participants between the ages of 20–79 years old were recruited through the Cambridge Biomedical Research Centre (prior to the COVID-19 pandemic) and were non-hospitalised, without SARS-CoV-2 infection, and had no neurological diagnoses. All participants provided written consent. Sex was not considered in the study design and the sex of participants was self-reported.

Participants were recruited into the COVID-Clinical Neuroscience Study (COVID-CNS) between October 2020 and October 2022 and either the participant or their next of kin consented in accordance with the ethically-approved NIHR Bioresource (East of England−Cambridge Central Research Ethics Committee (Ref 17/EE/0025; 22/EE/0230). The purpose of the study was to investigate patients who had been hospitalised with COVID-19 with or without neurological complications. These were defined by the following criteria: neurological disease onset within 6 weeks of acute SARS-CoV-2 infection and no evidence of other commonly associated causes, and diagnostic criteria previously described[48]. Participants were recruited both as in-patients and retrospectively after discharge. The diagnosis was reviewed and finalized by a multi-disciplinary Clinical Case Evaluation panel. In this study, there were COVID patients without neurological complications (COVID-controls) and COVID patients with neurological complications (Neuro-COVID cases) and these cases were stratified by diagnostic definitions of each type of neurological complication, very few had overlapping syndromes in this relatively small cohort and the Evaluation Panel were able to provide a primary diagnosis for all"[4]. Co-morbidities and known treatments are shown in Supplementary Table 7. Serum samples were collected at either the early (<6 weeks from COVID-19 positive test) or late convalescent (>6 weeks) phases. The samples were aliquoted, labelled with anonymised identifiers, and frozen immediately at −70 °C.

## Human brain injury markers measurements
Brain injury markers were measured in thawed sera using a Quanterix Simoa kit run on an automated HD-X Analyser according to the manufacturer's protocol (Quanterix, Billerica, MA, USA, Neurology 4-Plex B Advantage Kit, cat#103345). We assessed neurofilament light chain (NfL), Ubiquitin C-Terminal Hydrolase L1 (UCH-L1), total-Tau (tTau), and glial fibrillary acidic protein (GFAP) in sera diluted 1:4 and used the manufacturer's calibrators to calculate concentrations.

## Human serum cytokine measurements
Analytes in thawed sera were quantified using the BioRad human cytokine screening 48-plex kit (Cat# 12007283) following manufacturer's instructions on a Bioplex 200 using Manager software 6.2. This involved incubation of 1:4 diluted sera with antibody-coated magnetic beads, automated magnetic plate washing, incubating the beads with secondary detection antibodies, and adding streptavidin-PE. Standard curves of known protein concentrations were used to quantify analytes. Samples that were under the limit of detection were valued at the lowest detectable value adjusted for 1:4 dilution factor.

## Median-centred normalization of human serum cytokine measurements
To minimise any potential impact of any possible variation in sample storage and transport, concentrations were median-centred and normalised for each participant, using established methodology[49–51]. The pg/mL of cytokines were log-transformed and the median per participant across all cytokines was calculated. The log-transformed median was subtracted from each log-transformed value to generate a normalized set.

## Protein microarray autoantibody profiling
Autoantibodies were measured from thawed sera as previously described in Needham et al.[14]. Briefly, a protein array of antigens (based on the HuProt™ (version 4.0) platform) was used to measure bound IgM and IgG from sera, using secondary antibodies with different fluorescent labels detected by a Tecan LS400 scanner and GenePix Pro v4 software. As developed in previous studies[14,52], antibody positivity was determined by measuring the median fluorescence intensity (MFI) of the four quadruplicate spots of each antigen. The MFI was then normalized to the MFI of all antigens for that patient's sample by dividing each value by the median MFI. Z-scores were obtained from these normalized values based on the distribution derived for each antigen from the healthy control cohort. A positive autoantibody 'hit' was defined as an antigen where $Z \geq 3$.

## Detection of antibodies by immunohistochemistry
Immunohistochemistry was performed on sagittal sections of female Wistar rat brains. Brains were removed, fixed in 4% paraformaldehyde (PFA) at 4 °C for 1 h, cryoprotected in 40% sucrose for 48 h, embedded in freezing medium and snap-frozen in isopentane chilled on dry ice. 10-μm-thick sections were cut and mounted on slides in a cryostat. A standard avidin-biotin peroxidase method was used, as reported previously[53,54], where thawed sera were diluted 1:200 in 5% normal goat serum and incubated at 4 °C overnight, and secondary biotinylated goat anti-human IgG Fc was diluted (1:500) and incubated at room temperature for 1 h. Finally, slides were counter-stained using cresyl violet.

## Detection of autoantibodies with cell-based assays
HEK293T cells were seeded on 96 well plates in DMEM + 10% FCS at 37 °C and 5% $CO_2$, transiently transfected with polyethylenimine with the relevant antigen-encoding plasmids GABA$_B$-R1 and GABA$_B$-R2 of the GABA$_B$ receptor, membrane tethered LGI1, CASPR2 and the NR1 subunit of the NMDA receptor, as described previously[55–57]. Thawed serum samples were incubated at 1:100 dilution for CASPR2 and GABAB receptor assays, and at 1:20 for LGI1 and NMDAR. After washing, cells were fixed with 4% PFA, washed again and incubated with unconjugated goat anti-human IgG Fc antibody, and donkey anti-goat IgG heavy and light chain Alexa Fluor 568 antibody. Cells were co-stained with DAPI.

## Statistical analyses
Prism software (version 9.4.1, GraphPad Software Inc.) was used for graph generation and statistical analysis. The Shapiro-Wilk normality test used to check the normality of the distribution. Individual data points, median lines, and first and third quartiles are shown on box and whisker plots and violin plots with minimum and maximum points as error bars. Heatmaps, volcano plots and Chord diagrams were made using R studio (version 4.1.1 RStudio, PBC). The 2D cytokine network analyses were created using the qgraph package in R software and matrices differences were assessed by Steiger test[58]. Univariate analyses were conducted to test for differences between two groups. Differences between two normally distributed groups were tested using the paired or unpaired Student's t test as appropriate. The difference between two non-normally distributed groups

was tested using Mann–Whitney U test. Volcano plots used multiple Mann–Whitney U tests with a false discovery rate set to 5%, and heatmaps show Pearson's correlations adjusted for a false discovery rate of 5%. Group comparisons were by Kruskal–Wallis test. Frequency differences of antibodies were measured by Fisher's exact tests. Proteins which were statistically significantly different in the COVID-positive controls (GCS = 15 or COVID groups, respectively) versus the GCS less than or equal to 14 or neurological cases by Mann-Whitney test ($p \leq 0.05$) were analysed with the KEGG (Kyoto Encyclopedia of Genes and Genomes) database. Pathway classifications from the KEGG map search results were ranked by highest number of mapped candidates and exported in the KGML format using R package clusterProfiler. $p \leq 0.05$ was considered statistically significant.

### Reporting summary

Further information on research design is available in the Nature Portfolio Reporting Summary linked to this article.

## Data availability

The individual-level data from these studies is not publicly available to main confidentiality. Data generated by the ISARIC4C consortium is available for collaborative analysis projects through an independent data and materials access committee at isaric4c.net/sample_access. Data and samples from the COVID-Clinical Neuroscience Study are available through collaborative research by application through the NIHR bioresource at https://bioresource.nihr.ac.uk/using-our-bioresource/apply-for-bioresource-data-access/. Brain injury marker and immune mediator data are present in the paper and in the source data file. Source data are provided with this paper.

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

## Acknowledgements

We thank the patients and their loved ones who volunteered to contribute to these studies at one of the most difficult times in their lives, and the research staff in every hospital who recruited patients at personal risk under challenging conditions. This research was funded by the National Institute for Health and Care Research (NIHR) (CO-CIN-01) and jointly by NIHR and UK Research and Innovation (CV220-169, MC_PC_19059). B.D.M. is supported by the UKRI/MRC (MR/V03605X/1), the MRC/UKRI (MR/V007181/1), MRC (MR/T028750/1) and Wellcome (ISSF201902/3). C.D. is supported by MRC (MC_PC_19044). We would like to thank the University of Liverpool GCP laboratory facility team for Luminex assistance and the Liverpool University Biobank team for all their help, especially Dr. Victoria Shaw, Lara Lavelle-Langham, and Sue Holden. We would like to acknowledge the Liverpool Experimental Cancer Medicine Centre for providing infrastructure support for this research (Grant Reference: C18616/A25153). We acknowledge the Liverpool Centre for Cell Imaging (CCI) for provision of imaging equipment (Dragonfly confocal microscope) and excellent technical assistance (BBSRC grant number BB/R01390X/1). Tom Solomon is supported by The Pandemic Institute and the NIHR Health Protection Research Unit (HPRU) in Emerging and Zoonotic Infections at University of Liverpool. D.K.M. and E.N. are supported by the NIHR Cambridge Biomedical Centre and by NIHR funding to the NIHR BioResource (RG94028 and RG85445), and by funding from Brain Research UK 201819-20. We thank NIHR BioResource volunteers for their participation, and gratefully acknowledge NIHR BioResource centres, NHS Trusts and staff for their contribution. We thank the National Institute for Health and Care Research, NHS Blood and Transplant, and Health Data Research UK as part of the Digital Innovation Hub Programme. Support for title page creation and format was provided by AuthorArranger, a tool developed at the National Cancer Institute. The authors would like to acknowledge the eDRIS team (Public Health Scotland) for their support in obtaining approvals, the provisioning and linking of data and facilitating access to the National Safe Haven. The views expressed are those of the author(s) and not necessarily those of the UKRI, NHS, the NIHR or the Department of Health and Social Care.

## Author contributions

B.D.M., C.D., E.J.N., K.T., R.W., Y.H., G.K.W., C.C., J. Cavanagh, S.R.I., A.V., L.S.T. and D.K.M. designed, analysed data, interpreted experiments, and wrote the manuscript. T.S., J.P.S., G.B., M.G., J.-C.S., A.J.C., M.A.E. and A.P. provided scientific orientation and critically reviewed the manuscript. K. Stirrups, N.K., J.R.B., and P.F.C. provided oversight of the COVID-Clinical Neuroscience Study. J.K.B., P.J.M.O., and M.G.S. led the ISARIC4C study and provided oversight of the manuscript. G.S., A.G. and A.-C.C. analysed data. S.A.B., J. Clark, P.S., K. Subramaniam,

M.H., C.H. and F.N.E. performed preliminary feasibility experiments. C.D., R.D., A.R., E.N., M.L., S.E., A.F. and H.F. performed experiments and analysed the data. E.T. managed the collection of the patients' samples. ISARIC4C and COVID-CNS consortia recruited the patients for the study.

## Competing interests

T.S. is the Director of The Pandemic Institute which has received funding from Innova and CSL Seqirus and Aviva and DAM Health. T.S. was an advisor to the GSK Ebola Vaccine programme and the Siemens Diagnostic Programme. T.S. Chaired the Siemens Healthineers Clinical Advisory Board. T.S. Co-Chaired the WHO Neuro-COVID task force and sat on the UK Government Advisory Committee on Dangerous Pathogens, and the Medicines and Healthcare Products Regulatory Agency (MHRA) Expert Working Group on Covid-19 vaccines. T.S. Advised to the UK COVID-19 Therapeutics Advisory Panel (UK-TAP). T.S. was a Member of COVID-19 Vaccines Benefit Risk Expert Working Group for the Commission on Human Medicines (CHM) committee of the Medicines and Healthcare products Regulatory Agency (MHRA). T.S. has been a member of the Encephalitis Society since 1998 and President of the Encephalitis Society since 2019.

## Additional information

Benedict D. Michael [1,2,3,82]✉, Cordelia Dunai [1,2,82], Edward J. Needham [4,5], Kukatharmini Tharmaratnam [6], Robyn Williams [7,8], Yun Huang [1], Sarah A. Boardman [1], Jordan J. Clark [9,10,11], Parul Sharma [12], Krishanthi Subramaniam [12], Greta K. Wood [1], Ceryce Collie [1], Richard Digby [5], Alexander Ren [5], Emma Norton [5], Maya Leibowitz [5], Soraya Ebrahimi [5], Andrew Fower [7], Hannah Fox [7], Esteban Tato [13,14], Mark A. Ellul [1,3], Geraint Sunderland [1], Marie Held [15], Claire Hetherington [1], Franklyn N. Egbe [1], Alish Palmos [13,14], Kathy Stirrups [16,17], Alexander Grundmann [18,19], Anne-Cecile Chiollaz [20], Jean-Charles Sanchez [20], James P. Stewart [12], Michael Griffiths [1], Tom Solomon [1,2,3,21], Gerome Breen [13,14], Alasdair J. Coles [4], Nathalie Kingston [16,22], John R. Bradley [16,23], Patrick F. Chinnery [4,16], Jonathan Cavanagh [24], Sarosh R. Irani [7,8], Angela Vincent [25], J. Kenneth Baillie [26,27], Peter J. Openshaw [28,29], Malcolm G. Semple [1,2,30], ISARIC4C Investigators*, COVID-CNS Consortium*, Leonie S. Taams [31,83] & David K. Menon [5,83]

[1]Clinical Infection, Microbiology, and Immunology, Institute of Infection, Veterinary and Ecological Sciences, University of Liverpool, Liverpool L69 7BE, UK. [2]NIHR Health Protection Research Unit (HPRU) in Emerging and Zoonotic Infections at University of Liverpool, Liverpool L69 7BE, UK. [3]The Walton Centre NHS Foundation Trust, Liverpool L9 7BB, UK. [4]Department of Clinical Neurosciences, University of Cambridge, Cambridge CB2 0QQ, UK. [5]Division of Anaesthesia, Department of Medicine, University of Cambridge, Cambridge CB2 0QQ, UK. [6]Health Data Science, Institute of Population Health, University of Liverpool, Liverpool L69 3GF, UK. [7]Oxford Autoimmune Neurology Group, Nuffield Department of Clinical Neurosciences, University of Oxford, Oxford OX3 9DU, UK. [8]Departments of Neurology and Neuroscience, Mayo Clinic, Jacksonville, FL 32224, USA. [9]University of Liverpool, Liverpool L69 7BE, UK. [10]Department of Microbiology, Icahn School of Medicine, Mount Sinai, NY 10029, USA. [11]Center for Vaccine Research and Pandemic Preparedness (C-VARPP), Icahn School of Medicine, Mount Sinai, NY 10029, USA. [12]Infection Biology & Microbiomes, Institute of Infection, Veterinary and Ecological Sciences, University of Liverpool, Liverpool L3 5RF, UK. [13]Social, Genetic and Developmental Psychiatry Centre, Institute of Psychiatry, Psychology & Neuroscience, King's College London, London SE5 8AF, UK. [14]NIHR Maudsley Biomedical Research Centre, King's College London, London SE5 8AF, UK. [15]Centre for Cell Imaging, Liverpool Shared Research Facilities, Faculty of Health and Life Sciences, University of Liverpool, Liverpool L69 7ZB, UK. [16]NIHR BioResource, Cambridge University Hospitals NHS Foundation, Cambridge CB2 0QQ, UK. [17]Department of Haematology, University of Cambridge, Cambridge CB2 0QQ, UK. [18]Clinical Neurosciences, Clinical and Experimental Science, Faculty of Medicine, University of Southampton, Southampton SO17 1BF, UK. [19]Department of Neurology, Wessex Neurological Centre, University Hospital Southampton NHS Foundation Trust, Southampton SO16 6YD, UK. [20]Département de médecine interne des spécialités (DEMED), University of Geneva, Geneva CH-1211, Switzerland. [21]The Pandemic Institute, Liverpool L7 3FA, UK. [22]University of Cambridge, Cambridge CB2 0QQ, UK. [23]Department of Medicine, School of Clinical Medicine, University of Cambridge, Cambridge CB2 0QQ, UK. [24]Centre for Immunology, School of Infection & Immunity, College of Medical, Veterinary & Life Sciences, University of Glasgow, Glasgow G12 8TA, UK. [25]Nuffield Department of Clinical

Neurosciences, University of Oxford, Oxford OX3 9DU, UK. [26]Roslin Institute, University of Edinburgh, Edinburgh EH25 9RG, UK. [27]Intensive Care Unit, Royal Infirmary of Edinburgh, Edinburgh EH10 5HF, UK. [28]National Heart and Lung Institute, Imperial College London, London SW7 2BX, UK. [29]Imperial College Healthcare NHS Trust, London W2 1NY, UK. [30]Respiratory Unit, Alder Hey Children's Hospital NHS Foundation Trust, Liverpool L14 5AB, UK. [31]Centre for Inflammation Biology and Cancer Immunology, King's College London, London SE1 9RT, UK. [82]These authors contributed equally: Benedict D. Michael, Cordelia Dunai. [83]These authors jointly supervised this work: Leonie S. Taams, David K. Menon. *Lists of authors and their affiliations appear at the end of the paper. ✉e-mail: benmic@liverpool.ac.uk

## ISARIC4C Investigators

J. Kenneth Baillie[32], Peter J. Openshaw[33], Malcolm G. Semple[34], Beatrice Alex[32], Petros Andrikopoulos[33], Benjamin Bach[32], Wendy S. Barclay[33], Debby Bogaert[32], Meera Chand[35], Kanta Chechi[33], Graham S. Cooke[33], Ana da Silva[36], Thushan de Silva[37], Annemarie B. Docherty[32], Gonçalo dos Santos[33], Marc-Emmanuel Dumas[33], Jake Dunning[35], Tom Fletcher[38], Christopher A. Green[39], William Greenhalf[34], Julian L. Griffin[33], Rishi K. Gupta[40], Ewen M. Harrison[32], Antonia Y. Wai[36], Karl Holden[34], Peter W. Horby[41], Samreen Ijaz[35], Saye Khoo[34], Paul Klenerman[41], Andrew Law[32], Matthew R. Lewis[33], Sonia Liggi[33], Wei S. Lim[42], Lynn Maslen[33], Alexander J. Mentzer[43], Laura Merson[41], Alison M. Meynert[32], Shona C. Moore[34], Mahdad Noursadeghi[40], Michael Olanipekun[33], Anthonia Osagie[33], Massimo Palmarini[36], Carlo Palmieri[34], William A. Paxton[34], Georgios Pollakis[34], Nicholas Price[44], Andrew Rambaut[32], David L. Robertson[36], Clark D. Russell[32], Vanessa Sancho-Shimizu[33], Caroline J. Sands[33], Janet T. Scott[36], Louise Sigfrid[41], Tom Solomon[34], Shiranee Sriskandan[33], David Stuart[41], Charlotte Summers[45], Olivia V. Swann[32], Zoltan Takats[33], Panteleimon Takis[33], Richard S. Tedder[35], A. A. R. Thompson[37], Emma C. Thomson[36], Ryan S. Thwaites[33], Lance C. Turtle[34], Maria Zambon[35], Thomas M. Drake[32], Cameron J. Fairfield[32], Stephen R. Knight[32], Kenneth A. Mclean[32], Derek Murphy[32], Lisa Norman[32], Riinu Pius[32], Catherine A. Shaw[32], Marie Connor[34], Jo Dalton[34], Carrol Gamble[34], Michelle Girvan[34], Sophie Halpin[34], Janet Harrison[34], Clare Jackson[34], James Lee[41], Laura Marsh[34], Daniel Plotkin[41], Stephanie Roberts[34], Egle Saviciute[34], Sara Clohisey[32], Ross Hendry[32], Susan Knight[46], Eva Lahnsteiner[47], Gary Leeming[48], Lucy Norris[32], James Scott-Brown[32], Sarah Tait[46], Murray Wham[32], Richard Clark[47], Audrey Coutts[47], Lorna Donnelly[47], Angie Fawkes[47], Tammy Gilchrist[47], Katarzyna Hafezi[47], Louise MacGillivray[47], Alan Maclean[47], Sarah McCafferty[47], Kirstie Morrice[47], Lee Murphy[47], Nicola Wrobel[47], Gail Carson[41], Kayode Adeniji[47], Daniel Agranoff[47], Ken Agwuh[47], Dhiraj Ail[47], Erin L. Aldera[47], Ana Alegria[47], Sam Allen[47], Brian Angus[47], Abdul Ashish[47], Dougal Atkinson[47], Shahedal Bari[47], Gavin Barlow[47], Stella Barnass[47], Nicholas Barrett[47], Christopher Bassford[47], Sneha Basude[47], David Baxter[47], Michael Beadsworth[47], Jolanta Bernatoniene[47], John Berridge[47], Colin Berry[47], Nicola Best[47], Pieter Bothma[47], Robin Brittain-Long[47], Naomi Bulteel[47], Tom Burden[47], Andrew Burtenshaw[47], Vikki Caruth[47], David Chadwick[47], Duncan Chambler[47], Nigel Chee[47], Jenny Child[47], Srikanth Chukkambotla[47], Tom Clark[47], Paul Collini[47], Catherine Cosgrove[47], Jason Cupitt[47], Maria-Teresa Cutino-Moguel[47], Paul Dark[47], Chris Dawson[47], Samir Dervisevic[47], Phil Donnison[47], Sam Douthwaite[47], Andrew Drummond[47], Ingrid DuRand[47], Ahilanadan Dushianthan[47], Tristan Dyer[47], Cariad Evans[47], Chi Eziefula[47], Chrisopher Fegan[47], Adam Finn[47], Duncan Fullerton[47], Sanjeev Garg[47], Atul Garg[47], Effrossyni Gkrania-Klotsas[47], Jo Godden[47], Arthur Goldsmith[47], Clive Graham[47], Tassos Grammatikopoulos[49], Elaine Hardy[47], Stuart Hartshorn[47], Daniel Harvey[47], Peter Havalda[47], Daniel B. Hawcutt[47], Maria Hobrok[47], Luke Hodgson[47], Anil Hormis[47], Joanne Howard[47], Michael Jacobs[47], Susan Jain[47], Paul Jennings[47], Agilan Kaliappan[47], Vidya Kasipandian[47], Stephen Kegg[47], Michael Kelsey[47], Jason Kendall[47], Caroline Kerrison[47], Ian Kerslake[47], Oliver Koch[47], Gouri Koduri[47], George Koshy[47], Shondipon Laha[47], Steven Laird[47], Susan Larkin[47], Tamas Leiner[47], Patrick Lillie[47], James Limb[47], Vanessa Linnett[47], Jeff Little[47], Mark Lyttle[47], Michael MacMahon[47], Emily MacNaughton[47], Ravish Mankregod[47], Huw Masson[47], Elijah Matovu[47], Katherine McCullough[47], Ruth McEwen[47], Manjula Meda[47], Gary Mills[47], Jane Minton[47], Kavya Mohandas[47], Quen Mok[47], James Moon[47], Elinoor Moore[47], Patrick Morgan[47], Craig Morris[47], Katherine Mortimore[47], Samuel Moses[47], Mbiye Mpenge[47], Rohinton Mulla[47], Michael Murphy[47], Thapas Nagarajan[47], Megan Nagel[47], Mark Nelson[47], Lillian Norris[47], Matthew K. O'Shea[47], Marlies Ostermann[44], Igor Otahal[47], Mark Pais[47], Selva Panchatsharam[47], Danai Papakonstantinou[47], Padmasayee Papineni[47], Hassan Paraiso[47], Brij Patel[47], Natalie Pattison[47], Justin Pepperell[47], Mark Peters[47], Mandeep Phull[47], Stefania Pintus[47], Tim Planche[47], Frank Post[47], David Price[47], Rachel Prout[47], Nikolas Rae[47], Henrik Reschreiter[47], Tim Reynolds[47], Neil Richardson[47], Mark Roberts[47], Devender Roberts[47], Alistair Rose[47], Guy Rousseau[47], Bobby Ruge[47], Brendan Ryan[47], Taranprit Saluja[47], Matthias L. Schmid[47], Aarti Shah[47], Manu Shankar-Hari[47], Prad Shanmuga[47], Anil Sharma[47], Anna Shawcross[47], Jagtur S. Pooni[47], Jeremy Sizer[47], Richard Smith[47], Catherine Snelson[47], Nick Spittle[47], Nikki Staines[47], Tom Stambach[47], Richard Stewart[47], Pradeep Subudhi[47], Tamas Szakmany[47], Kate Tatham[47], Jo Thomas[47], Chris Thompson[47], Robert Thompson[47], Ascanio Tridente[47], Darell Tupper-Carey[47], Mary Twagira[47], Nick Vallotton[47], Rama Vancheeswaran[47], Rachel Vincent[47], Lisa Vincent-Smith[47], Shico Visuvanathan[47], Alan Vuylsteke[47], Sam Waddy[47], Rachel Wake[47],

Andrew Walden[47], Ingeborg Welters[47], Tony Whitehouse[47], Paul Whittaker[47], Ashley Whittington[47], Meme Wijesinghe[47], Martin Williams[47], Lawrence Wilson[47], Stephen Winchester[47], Martin Wiselka[47], Adam Wolverson[47], Daniel G. Wootton[47], Andrew Workman[47], Bryan Yates[47], Peter Young[47], Sarah E. McDonald[36], Victoria Shaw[34], Katie A. Ahmed[47], Jane A. Armstrong[47], Milton Ashworth[47], Innocent G. Asiimwe[47], Siddharth Bakshi[47], Samantha L. Barlow[47], Laura Booth[47], Benjamin Brennan[47], Katie Bullock[47], Nicola Carlucci[47], Emily Cass[47], Benjamin W. Catterall[47], Jordan J. Clark[47], Emily A. Clarke[47], Sarah Cole[47], Louise Cooper[47], Helen Cox[47], Christopher Davis[47], Oslem Dincarslan[47], Alejandra D. Carracedo[47], Chris Dunn[47], Philip Dyer[47], Angela Elliott[47], Anthony Evans[47], Lorna Finch[47], Lewis W. Fisher[47], Lisa Flaherty[47], Terry Foster[47], Isabel Garcia-Dorival[47], Philip Gunning[47], Catherine Hartley[47], Anthony Holmes[47], Rebecca L. Jensen[47], Christopher B. Jones[47], Trevor R. Jones[47], Shadia Khandaker[47], Katharine King[47], Robyn T. Kiy[47], Chrysa Koukorava[47], Annette Lake[47], Suzannah Lant[47], Diane Latawiec[47], Lara Lavelle-Langham[47], Daniella Lefteri[47], Lauren Lett[47], Lucia A. Livoti[47], Maria Mancini[47], Hannah Massey[47], Nicole Maziere[47], Sarah McDonald[47], Laurence McEvoy[47], John McLauchlan[47], Soeren Metelmann[47], Nahida S. Miah[47], Joanna Middleton[47], Joyce Mitchell[47], Ellen G. Murphy[47], Rebekah Penrice-Randal[47], Jack Pilgrim[47], Tessa Prince[47], Will Reynolds[47], P. M. Ridley[47], Debby Sales[47], Victoria E. Shaw[47], Rebecca K. Shears[47], Benjamin Small[47], Krishanthi S. Subramaniam[47], Agnieska Szemiel[47], Aislynn Taggart[47], Jolanta Tanianis-Hughes[47], Jordan Thomas[47], Erwan Trochu[47], Libby v. Tonder[47], Eve Wilcock[47], J. E. Zhang[47], Seán Keating[50], Cara Donegan[34], Rebecca G. Spencer[34], Chloe Donohue[34], Fiona Griffiths[51], Hayley Hardwick[34] & Wilna Oosthuyzen[32]

[32]University of Edinburgh, Edinburgh, UK. [33]Imperial College London, London, UK. [34]University of Liverpool, Liverpool, UK. [35]Public Health England, London, UK. [36]MRC-University of Glasgow Centre for Virus Research, Glasgow, UK. [37]University of Sheffield, Sheffield, UK. [38]Liverpool School of Tropical Medicine, Liverpool, UK. [39]University of Birmingham, Birmingham, UK. [40]University College London, London, UK. [41]University of Oxford, Oxford, UK. [42]Nottingham University Hospitals NHS Trust, Nottingham, UK. [43]John Radcliffe Hospital, Oxford, UK. [44]King's College London, London, UK. [45]University of Cambridge, Cambridge, UK. [46]Public Health Scotland, Edinburgh, UK. [47]ISARIC4C Investigators, Liverpool, UK. [48]University of Manchester, Manchester, UK. [49]King's College Hospital, London, UK. [50]Royal Infirmary Edinburgh, Edinburgh, UK. [51]Roslin Institute, Edinburgh, UK.

## COVID-CNS Consortium

Adam Hampshire[33], Adam Sieradzki[52], Adam W. Seed[53], Afagh Garjani[54], Akshay Nair[44], Alaisdair Coles[45], Alan Carson[55], Alastair Darby[34], Alex Berry[40], Alex Dregan[44], Alexander Grundmann[56], Alish Palmos[44], Ammar Al-Chalabi[44], Andrew M. McIntosh[32], Angela E. Holland[57], Angela Roberts[45], Angela Vincent[41], Annalena Venneri[37], Anthony S. David[40], Arina Tamborska[34], Arvind Patel[58], Ava Easton[34], Benedict D. Michael[34], Bethan Blackledge[59], Bethany Facer[34], Bhagteshwar Singh[34], Brendan Sargent[34], Ceryce Collie[34], Charles Leek[34], Cherie Armour[60], Christopher M. Morris[61], Christopher M. Allen[54], Ciaran Mulholland[60], Claire L. MacIver[60], Cordelia Dunai[34], Craig J. Smith[48], Daniel J. van[44], Daniel Madarshahian[63], David Christmas[64], David Cousins[61], David K. Menon[45], David M. Christmas[45], David P. Breen[32], Dina Monssen[44], Edward Bullmore[45], Edward Needham[45], Emily McGlinchey[60], Emma Thomson[58], Eugene Duff[41], Eva M. Hodel[34], Ewan Harrison[45], Fernando Zelaya[44], Gabriella Lewis[65], Gavin McDonnell[66], Gerome Breen[44], Greta K. Wood[34], Guy B. Williams[45], C. Hannah[34], Henry C. Rogers[44], Ian Galea[56], Jacqueline Smith[67], Jade D. Harris[59], James B. Lilleker[68], Jay Amin[56], John P. Aggleton[62], John R. Bradley[45], John-Paul Taylor[61], Jonathan Cavanagh[58], Jonathan R. Coleman[44], Jonathan Underwood[62], Judith Breuer[40], Julian Hiscox[34], Karla Miller[41], Katherine C. Dodd[48], Kiran Glen[44], Laura Benjamin[40], Leonie Taams[44], Lily George[44], Marc Hardwick[56], Mark R. Baker[61], Marlies Ostermann[43], Masud Husain[41], Matthew Butler[44], Matthew Hotopf[44], Matthew R. Broome[39], Merna Samuel[34], Michael Griffiths[34], Michael P. Lunn[40], Michael S. Zandi[40], Monika Hartmann[69], Nadine Cossette[70], Naomi Martin[44], Nathalie Nicholas[71], Neil A. Harrison[62], Neil Basu[58], Neil Harrison[62], Nicholas Davies[33], Nicholas Wood[40], Nikos Evangelou[54], Obioma Orazulume[40], Pamela J. Shaw[37], Parisa Mansoori[72], Paul J. Harrison[41], Peter Jezzard[41], Peter M. Fernandes[32], Rachel Upthegrove[73], Rahul Batra[44], Rebecca Gregory[74], Rhys H. Thomas[75], Richard Bethlehem[45], Richard Francis[76], Ronan O'Malley[77], Rustam A. Salman[78], Ryan McIlwaine[60], Sandar Kyaw[79], Sarosh Irani[41], Savini Gunatilake[80], Scott Semple[32], Shahd H. Hamid[34], Sharon Peacock[45], Silvia Rota[44], Simon Keller[34], Sophie Pendered[34], Suzanne Barrett[81], Stella Hughes[66], Stella-Maria Paddick[61], Stephen J. Sawcer[45], Stephen Smith[41], Steven Williams[44], Sui H. Wong[44], Sylviane Defres[38], Thomas Jackson[73], Thomas M. Jenkins[37], Thomas Pollak[44], Timothy Nicholson[44], Tom Solomon[33], Tonny Veenith[39], Victoria Grimbly[34] & Virginia Newcombe[45]

[52]COVID-CNS Consortium, York, UK. [53]Liverpool University Hospitals NHS Foundation Trust, Liverpool, UK. [54]University of Nottingham, Nottingham, UK. [55]Edinburgh University, Edinburgh, UK. [56]University of Southampton, Southampton, UK. [57]Nottingham University Hospital, Nottingham, UK. [58]University of Glasgow, Glasgow, UK. [59]Salford Royal Foundation Trust, Manchester, UK. [60]Queens University Belfast, Belfast, UK. [61]Newcastle University, Newcastle, UK. [62]Cardiff University, Cardiff, UK. [63]Sheffield Institute for Translational Neuroscience, Sheffield, UK. [64]Dundee University, Dundee, UK. [65]South London and Maudsley NHS Foundation Trust, London, UK. [66]Belfast Health and Social Care Trust, Belfast, UK. [67]COVID-CNS Consortium, Edinburgh, UK. [68]The University

of Manchester, Manchester, UK. [69]Kings College London, London, UK. [70]Royal Infirmary of Edinburgh, Edinburgh, UK. [71]Aintree University Hospital, Liverpool, UK. [72]National Institute for Health Research (NIHR) Bioresource, London, UK. [73]Birmingham University, Birmingham, UK. [74]Sheffield Teaching Hospitals NHS Foundation Trust, Sheffield, UK. [75]Translational and Clinical Research, Newcastle, UK. [76]The Stroke Association, London, UK. [77]The University of Sheffield, Sheffield, UK. [78]The University of Edinburgh, Edinburgh, UK. [79]Institute of Mental health, Nottingham, UK. [80]Royal Stoke University Hospital, Stoke on Trent, UK. [81]Northern Health and Social Care Trust, Belfast, UK.

