## [Peer Review File · Nature Communications]

Para-infectious brain injury in COVID-19 persists at follow-up despite attenuated cytokine and autoantibody responsesReviewers' Comments:

Reviewer #1:

Remarks to the Author:

This is an interesting and very important manuscript relating neurologically relevant biomarkers to the acute and more chronic neurologic manifestations of COVID-19 infection. The particularly interesting results include the persistent elevation of NfL and GFAP in those who are convalescent from acute neurological COVID complications, correlation between activation of the innate immune system and elevated CNS biomarkers. The human biomarker studies are well described and detailed. However, this reviewer has some questions about the mouse studies:

1. Why were heterozygous hACE2-transgenic C57BL/6 mice used versus homozygous? Several concerns here- variability in hACE2 expression from mouse to mouse AND, as this mouse has a keratin promoter for human ACE2 receptor expression, the ACE2 is expressed in many cell types that it would not normally be expressed in. Also, the ability of the hACE2 receptor to affect biologically relevant signaling in mouse cells is likely very problematic. This mouse, therefore, is excellent for asking whether the virus can infect at a certain inoculum, and whether any experimental treatment might impact that yes or no infection question, but NOT very good at investigating secondary pathogenesis resulting from that infection, ESPECIALLY in the brain, where hACE2 is more present in these mice in the brain than ACE2 is otherwise present in WT mice. As the alpha variant of SARS-CoV-2 and omicron have been reported to be able to infect WT mice, either would have been a much better choice when specifically looking for neurological aspects of COVID-19 in a mouse model. Likewise, there are published mouse adapted SARS-CoV-2 that would also work well to examine this question in WT mice. Really, the mouse studies described in this manuscript seem like a separate, and not really related study from the human clinical data (which IS very interesting and important) and do little to support the human biomarker assessments, especially given the low inoculum (patients clearly did not have this, especially the ones with acute neurological issues and / or hospitalized) and the 5 day time course used (no convalescent component to go along with the human convalescent component). Another minor concern- is the virus used in the mouse studies likely relatable to the virus that was likely to have infected the human patients at the time that the blood was collected from them? As different variants are known to have different impacts on the brain or other organs, this point should be discussed.

Reviewer #2:

Remarks to the Author:

In this manuscript the authors investigate a cohort of hospitalized patients with neurological complications for evidence of markers of neuronal injury, inflammation and autoantibodies during the acute and convalescent phase. This is an important study, the strengths of which include a control group, large sample size and an attempt to replicate the findings in an animal model. An interesting observation is the increase in makers of neuronal injury in patients with COVID but without any neurological complications.

1. It is unclear how well the controls are matched. They have a very broad age range and the mean age is 10 years lower than the Neuro-COVID group. This is important since levels of markers of neuronal injury are age dependent.
2. The observations with autoantibodies are hard to interpret. It does seem that there might be polyclonal B cell activation with low level autoantibodies to a number of antigens in the acute phase which is to be expected. However, they seem to have a subset of patients with antibodies to HLA antigens which remains unexplained and they also found some patients with antibodies against neuronal antigens on immunostaining of mouse brain tissue but these antibodies have not been further characterized. It is not clear if the protein array has membrane proteins on not. Since clinically significant autoantibodies are most often against membrane antigens, it would be hard to draw any conclusions about autoantibodies in this cohort without further investigation.
3. Extended data table 2: It would be useful to better define what is meant by encephalopathy and

encephalitis. Encephalopathy is not typically thought to be an inflammatory condition. Also is the demyelination due an inflammatory process. It might be better to use the terms ADEM, TM, AHE or MISC if that is how these patients presented.

4. It is not surprising that patients with acute neurological illnesses would have elevated levels of markers of neuronal injury in the blood. What is a bit unusual is that in the early convalescent phase they found no differences between those with and without neurological complications (lines 275-277). But no explanation is provided for this observation.

5. Lines 285-287: The correlation between tTau and cytokines is interesting but no explanation is provided as to why there is no correlation with NfL levels since they both represent axonal damage.

6. Line 401: It is stated that the samples were heat inactivated. Would it be possible to provide more details on how the samples were handled. Any repeated freeze thaws? To what temperature were the samples heated and for how long?

7. Lines 445-449: Some diagnostic criteria were used but no references or details are provided. They have tried to categorize patients into a few neurological syndromes. Were there no patients with overlapping syndromes?

8. The mouse model is poorly characterized. For example, in this study a sizable number of patients had cerebrovascular disease. It would be important to characterize the vascular pathology in the brains of these mice.

9. Similarly, immunostaining for neuronal markers to look for axonal or cellular damage would be important if the goal is to recapitulate the clinical observations in these animals.

10. Extended data figure 6: the colors in panels d-f are hard to see. This could very well be due to poor resolution in the pdf files. If the colors can be enhanced, it would be helpful.

11. It is not clear why the mice were not allowed to live beyond 5 days. Did they develop severe systemic or pulmonary dysfunction? This is too short a time frame to look for the types of neurological syndromes seen in the patient population in this study. I wonder if this model adds much and may be this part can be removed and published separately when it is fully characterized. A similar model has already been described previously and they can just reference that model (Fernandez-Castaneda et al., Cell 2022).

Avindra Nath

Reviewer #3:

Remarks to the Author:

This is a retrospective observational study measuring blood biomarkers of cellular injury to the nervous system and inflammatory molecules from COVID-19 patients with and without neurologic dysfunction compared to healthy volunteer control sera.

The authors utilized two different cohorts for measurements of serum inflammatory mediators and biomarkers of nervous system cellular injury. One cohort of subjects had a blood sample obtained within 11 days of admission for COVID-19 (ISARIC). This acute illness cohort was sub-divided into normal neurologic function (GCS 15) or neurologic dysfunction (GCS 14 or less). The second cohort consisted of subjects previously diagnosed with COVID-19 and subsequently diagnosed with onset of a neurologic disease within 6 weeks of diagnosis (COVID-CNS). These subjects had serum samples drawn less than 6 weeks from admission (early convalescent) or > 6 weeks from admission (late convalescent).

A mouse model to simulate the neuropathology seen in human COVID-19 patients was also developed by the authors.

These data provide insights into some of the immunological changes that may mediate neurological dysfunction following COVID-19. However, the data are tempered by the comments below.

Overall, this manuscript is difficult to follow at times. Providing a focused hypothesis stating the objective of this study would provide readers context for why this study is important and why specific assays were utilized to address the hypothesis.

Methods:

Two separate patient cohorts impacts the continuity of this study. Furthermore, the characterization of

neurological dysfunction differs between the two cohorts. The ISARIC cohort neurologic dysfunction was defined as GCS 15 compared to equal to or less than 14. The COVID-CNS cohort provided greater granularity in defining specific neurological disorders. Therefore, the continuity of neurological dysfunction differs from cohort to cohort. The authors should provide a rationale for using two separate cohorts using different characterizations of neurological dysfunction.

The mouse SARS-CoV-2 infection studies used two different titers for infection (low vs high inoculation). What data supported the specific concentrations of virus used for infections? Is there any data correlating these titers to humans with COVID-19 with or without neurological dysfunction? Why were rat brains, as opposed to mouse, used for incubation with human sera? What is the data to support cross reactivity of human antibodies from COVID-19 patients with rat brain epitopes? Line 522 appears to have a place-holder for model details regarding the Leica confocal microscope.

Results:

Clinical data from both cohorts is very limited. The focus of this manuscript are inflammatory changes in COVID-19 patients with neurological dysfunction. However, there is no epidemiological data regarding premorbid conditions of the subjects enrolled, nor is there any data regarding what concurrent morbidities that may have contributed to a subjects alteration in GCS or neurological dysfunction. For example, how many subjects had hypoxia and/or hypercapnia associated with COVID-19, which may affect neurologic function. No data was presented regarding confounding medications that may affect neurological function, such as analgesia, sedation, antiseizure medications. Very importantly to this study's findings of elevated inflammatory mediators and autoantibodies, there are no reported data regarding the administration of immunosuppressive/immunomodulatory medications such as dexamethasone, remdesivir, or tocilizumab often used for the treatment of hospitalized COVID-19 patients. The authors briefly mention this in the discussion, but this really should be added to this manuscript due to potential significant effect on the regulation and function of immune mediators. It is difficult to accurately interpret the results of this study without knowing which subjects received immunosuppressive/immunomodulatory medications.

In figure 1 early and late convalescent were lumped together in subjects positive for COVID with or without neurologic disease. Were there differences between early versus late convalescent subjects? The autoantibody assays measured both IgM and IgG reactivity from ISARIC sera. These subjects had their blood drawn within first 11 days of admission. Please comment on how to tease out IgG autoantibodies associated with the acute COVID infection from prior illness or antigen exposure. What is the potential effect from prior environmental antigens?

Figure 3 states acute sera containing IgG antibodies against CNS proteins, however these assays utilized acute samples and measured IgG antibodies instead of IgM antibodies. The text in the results section states IgG and IgM antibodies were measure from sera. Please clarify.

Mice were infected with low vs high viral titers. Both groups showed viral replication in the brain, with the lower inoculated mice with less viral replication. Why were only the low inoculated mice data shown in figure 5? What did the high inoculated mice show in regards to inflammatory mediators compared to low inoculated mice?

Discussion:

Regarding lines 291 and 292 in the discussion how do we know that IgG autoantibodies were associated with SARS-CoV2 infection? This could be due to a prior environmental antigen since this study is measuring IgG.

Line 304 of the discussion states "absence of viral replication in the brain parenchyma" however SARS-CoV-2 N1 transcript was detected in four of five brains of mice that had received high inoculum of SARS-CoV-2 and in six of nine that received low inoculum. Please clarify this statement with data presented in figure 4.

Cytokines were measured from serum samples and not from CSF. What are potential systemic effects these cytokines have directly on brain constituent cells and cerebrovasculature?

The discussion states the potential effect of injury to the cerebral vasculature in mediating

neurological dysfunction following COVID-19. Why was brain vascular histology or biomarkers of endothelial glycocalyx/blood-brain barrier degradation not included since these mice were infected with SARS-CoV-2? This is especially important in light of data from acutely infected subjects where IgM may mediate a role in neuronal dysfunction. IgM are pentamers with approximately molecular weight of 900 kDa, which would require BBB permeability in order to gain entry into the brain tissue.

**Point-by-point response letter for reviewers for *Nature Communications***
**resubmission of manuscript:**

Para-infectious brain injury in COVID-19 persists at follow-up despite attenuated cytokine
and autoantibody responses

All new text in manuscript is in red colour

**REVIEWER COMMENTS**

**Reviewer #1 (Remarks to the Author):**

This is an interesting and very important manuscript relating neurologically relevant
biomarkers to the acute and more chronic neurologic manifestations of COVID-19
infection. The particularly interesting results include the persistent elevation of NfL
and GFAP in those who are convalescent from acute neurological COVID
complications, correlation between activation of the innate immune system and
elevated CNS biomarkers. The human biomarker studies are well described and
detailed. However, this reviewer has some questions about the mouse studies:

1. Why were heterozygous hACE2-transgenic C57BL/6 mice used versus
homozygous? Several concerns here- variability in hACE2 expression from mouse to
mouse AND, as this mouse has a keratin promoter for human ACE2 receptor
expression, the ACE2 is expressed in many cell types that it would not normally be
expressed in. Also, the ability of the hACE2 receptor to affect biologically relevant
signaling in mouse cells is likely very problematic. This mouse, therefore, is excellent
for asking whether the virus can infect at a certain inoculum, and whether any
experimental treatment might impact that yes or no infection question, **but NOT very**
**good at investigating secondary pathogenesis resulting from that infection,**
**ESPECIALLY in the brain**, where hACE2 is more present in these mice in the brain
than ACE2 is otherwise present in WT mice. As the alpha variant of SARS-CoV-2
and omicron have been reported to be able to infect WT mice, either would have
been a much better choice when specifically looking for neurological aspects of
COVID-19 in a mouse model. Likewise, there are published mouse adapted SARS-
CoV-2 that would also work well to examine this question in WT mice.

Really, the **mouse studies described in this manuscript seem like a separate,**
**and not really related study** from the human clinical data (which IS very interesting
and important) and do little to support the human biomarker assessments, especially
given the low inoculum (patients clearly did not have this, especially the ones with
acute neurological issues and / or hospitalized) and the **5 day time course** used (no
convalescent component to go along with the human convalescent component).

Another minor concern- is the virus used in the mouse studies likely relatable to the
virus that was likely to have infected the human patients at the time that the blood
was collected from them? As different variants are known to have different impacts
on the brain or other organs, this point should be discussed.

Response: Thank you for these important points and feedback on the mouse model.
We agree that there are many caveats and limitations to extrapolating the findings
from the human ACE2 transgenic mice. As a result, we have removed all the mouse
model data (partly on the editor's recommendation) and now only present clinical
findings in this paper to focus on those analyses without making comparisons with
the mouse model.

**Reviewer #2 (Remarks to the Author):**

In this manuscript the authors investigate a cohort of hospitalized patients with
neurological complications for evidence of markers of neuronal injury, inflammation
and autoantibodies during the acute and convalescent phase. This is an important
study, the strengths of which include a control group, large sample size and an
attempt to replicate the findings in an animal model. An interesting observation is the
increase in makers of neuronal injury in patients with COVID but without any
neurological complications.

**1. It is unclear how well the controls are matched.** They have a very broad age
range and the mean age is 10 years lower that the Neuro-COVID group. This is
important since levels of markers of neuronal injury are age dependent.

Response: This is an important point as serum brain injury markers do increase with
age. We have conducted age-adjusted analysis and presented this in Supplementary
Data Figure 1a,b. The brain injury marker NfL remains significantly elevated in the
COVID-CNS (Neuro-COVID) cohort even when adjusted for age. This could be
related to the severity of the neurological complications observed in young
participants. This is addressed in lines 118-119:

"NfL remained significantly different in a multiple regression model adjusted for age
(Supplementary Data Fig 1a,b)."

**2. The observations with autoantibodies are hard to interpret.** It does seem that there
might be polyclonal B cell activation with low level autoantibodies to a number of
antigens in the acute phase which is to be expected. However, they seem to have a
subset of patients with antibodies to HLA antigens which remains unexplained and
they also found some patients with antibodies against neuronal antigens on
immunostaining of mouse brain tissue but these antibodies have not been further
characterized. It is not clear if the protein array has membrane proteins on not. Since
clinically significant autoantibodies are most often against membrane antigens, it
would be hard to draw any conclusions about autoantibodies in this cohort without
further investigation.

Response: We understand the Reviewer's concerns regarding the interpretation of
the antibodies that were measured by the HuProt microarray. Although it included a
number of neuronal cell membrane receptors, we were not able to confirm antibodies
binding to them by cell-based assays, suggesting that either the levels are too low to
detect with the routine clinical assays used, or that the antigens on the HuProt were

not conformational despite the manufacturers' intention (now shown in
Supplementary Figure 7). This adds to other experience from the Oxford lab that
those antibodies binding on microarrays seldom if ever bind to the native membrane
receptors. Moreover, none of the fluorescence scores of the HuProt antibodies were
high and only the frequencies of some of them were greater than the relevant
controls. Thus our interpretation remains that the binding seen is more indicative of
a general B cell activation, and certainly not exclusively directed at neuronal antigens
even in those patients with neurological symptoms.

Regarding the binding to brain tissue, that is always difficult to interpret since both
intracellular and membrane proteins are detected by that approach and the use of
fixed tissue also complicates interpretation. Experience suggests that non-specific
binding is common with this technique. The binding to the brainstem region was only
present in a small number of the COVID-CNS patients and did not discriminate
between them and the non-neurological participants, but does remain of interest
since it was more frequent in patients than in controls. This requires confirmation by
others and if it can be shown to be of clinical relevance, antigen discovery
approaches should be applied in order to discover whether the brainstem antibodies
are to cell-membrane proteins and potentially pathogenic. Those are outside the
remit of this study.

Finally, the reviewer questions the significance of the subset of patients who had
antibodies to HLA antigens on the HuProt microarray. These were more common in
the CNS patients and are thus intriguing but their significance is unclear. As
discussed, the HuProt microarray is highly sensitive and HLA antibodies are not so
uncommon in the general population, particularly in parous women or after blood
transfusions^{1,2}. It would be hard to draw any conclusions about autoantibodies in this
cohort without further investigation.

This is now discussed in the results and discussion (below).

Lines 210-217

"Binding to rat brain sections identified 42/185 (23%) of participants with strongly
positive immunohistochemical staining (eg. Fig. 4i) and overall, sera from the
COVID+ve ISARIC participants showed more frequent binding to brainstem regions
than control sera, but this did not relate to the GCS or neurological disease of the
participants (Fig. 4j, Supplementary Fig. 6). In addition, from 34 selected samples
tested via cell-based assays to examine for the presence of specific autoantibodies
(LGI1, CASPR2, NMDAR, GABAB receptor), only one bound to the extracellular
domain of the GABAB receptor (from the ISARIC cohort, Supplementary Fig. 7a,b),
as expected of a pathogenic autoantibody."

Lines 293-295

"The autoantibodies detected in COVID-19, as in other infections, could be through
molecular mimicry or bystander effects,³⁻⁶ but the lack of association of autoantibody
levels with markers of brain injury is evidence against a causal role for these
adaptive immune responses."

3. Supplementary data table 2: It would be useful to better define what is meant by encephalopathy and encephalitis. Encephalopathy is not typically thought to be an inflammatory condition. Also is the demyelination due an inflammatory process. It might be better to use the terms ADEM, TM, AHE or MISC if that is how these patients presented.

Response: The reviewer raises an important distinction between encephalitis and encephalopathy based on raised CSF white blood cell counts. To distinguish this group from CNS inflammation, encephalopathy have now been grouped with Central/other, in accordance with the COVID-19 criteria described by Ellul MA, et al. Lancet Neurol 2020⁷ (Supplementary Table 2).

For the demyelinating disorders, we have now reported the subclassifications in Supplementary Table 2).

4. It is not surprising that patients with acute neurological illnesses would have elevated levels of markers of neuronal injury in the blood. What is a bit unusual is that in the early convalescent phase they found no differences between those with and without neurological complications (lines 275-277). But no explanation is provided for this observation.

Response: This is a very interesting point, and this finding has been reported in many studies —patients with COVID-19 even without neurological complications have raised levels of brain injury markers. Our previous work has shown that the NfL and Tau correlate with COVID severity indicating non-specific damage occurring in the CNS⁸. This is now discussed in more depth in lines 75-79 and we discuss how CSF brain injury markers might be more correlated with neurological outcomes.

“The brain injury markers NfL and GFAP, and inflammatory cytokines were elevated in COVID-19 and scaled with severity²¹⁻²⁵; another study showed that baseline CSF NfL levels correlated with neurological outcomes at follow-up²⁶ but overall, the relationships between these biomarkers and neuropathology remains to be fully explored.”

The Reviewer is quite right that in the early convalescent samples there was only a trend towards greater elevation in brain injury markers (NfL and GFAP) between the NeuroCOVID and COVID groups as both were significantly higher than controls.

We acknowledge that this is does not reach statistical significance and that this may reflect the relatively small numbers at these time points.

In the revised manuscript we make it clear that the trend to elevated NfL and GFAP is most evident in late convalescent samples as both reach statistical significance as elevated in patients with NeuroCOVID vs controls, which is not present for COVID cases without neurological complications (Figure 1n,p).

5. Lines 285-287: The correlation between tTau and cytokines is interesting but no explanation is provided as to why there is no correlation with NfL levels since they both represent axonal damage.

Response: The reviewer is correct that NfL and tTau levels correlate as they both reflect axonal damage, however, the correlation of the cytokines did not reach significance when compared with NfL. This could be due to a few exceptionally high tTau values driving the correlation with the 8 cytokines (Supplementary Data Fig. 2c).

6. Line 401: It is stated that the samples were heat inactivated. Would it be possible to provide more details on how the samples were handled. Any repeated freeze thaws? To what temperature were the samples heated and for how long?

Response: This referred to the mouse sera (and not the clinical samples) which were heat-inactivated and thawed one or two times. This data has now been removed from the manuscript. The human sera were not heat-inactivated and went through one or two freeze-thaws.

7. Lines 445-449: Some diagnostic criteria were used but no references or details are provided. They have tried to categorize patients into a few neurological syndromes. Were there no patients with overlapping syndromes?

Response: Thank you, this is now clarified in the methods lines 369-371:

“These were defined by the following criteria: neurological disease onset within 6 weeks of acute SARS-CoV-2 infection and no evidence of other commonly associated causes, and diagnostic criteria previously described⁷.”

Lines 372-378

“The diagnosis was reviewed and finalized by a multi-disciplinary Clinical Case Evaluation panel. In this study there were COVID patients without neurological complications (COVID-controls) and COVID patients with neurological complications (Neuro-COVID cases) and these cases were stratified by diagnostic definitions of each type of neurological complication, very few had overlapping syndromes in this relatively small cohort and the Evaluation Panel were able to provide a primary diagnosis for all⁹”.

8. The mouse model is poorly characterized. For example, in this study a sizable number of patients had cerebrovascular disease. It would be important to characterize the vascular pathology in the brains of these mice.

Response: This is a very good point, but now outside the scope of this manuscript as the mouse model has been separated out.

9. Similarly, immunostaining for neuronal markers to look for axonal or cellular damage would be important if the goal is to recapitulate the clinical observations in these animals.

Response: At the advice of the Reviewers and Editors, the mouse data has now
been removed from this manuscript.

10. Supplementary data figure 6: the colors in panels d-f are hard to see. This could
very well be due to poor resolution in the pdf files. If the colors can be enhanced, it
would be helpful.

Response: Thank you for pointing this out. We will have better resolution for the
separate paper that will cover the mouse model.

11. It is not clear why the mice were not allowed to live beyond 5 days. Did they
develop severe systemic or pulmonary dysfunction? This is too short a time frame to
look for the types of neurological syndromes seen in the patient population in this
study. I wonder if this model adds much and may be this part can be removed and
published separately when it is fully characterized. A similar model has already been
described previously and they can just reference that model (Fernandez-Castaneda
et al., Cell 2022).

Response: The mice would have survived beyond 5 days as they had very mild
phenotype. This timepoint was used for comparison with other studies. We have
removed the mouse model data and we cite the relevant Fernandez-Castaneda et
al., Cell 2022 paper in the discussion lines 328-333

“A recent mouse study is particularly relevant to our work and involved assessment
of a mouse model that also lacked direct viral neural invasion by infecting mice that
were intratracheally transfected with human ACE2. This study reported increased
CXCL11 (eotaxin) in mouse serum and CSF that correlated with demyelination and
was recapitulated by giving CXCL11 intraperitoneally¹⁰; this was linked to clinical
studies that showed elevated CXCL11 in patients with brain fog¹⁰.”

**Reviewer #3 (Remarks to the Author):**

This is a retrospective observational study measuring blood biomarkers of cellular
injury to the nervous system and inflammatory molecules from COVID-19 patients
with and without neurologic dysfunction compared to healthy volunteer control sera.
The authors utilized two different cohorts for measurements of serum inflammatory
mediators and biomarkers of nervous system cellular injury. One cohort of subjects
had a blood sample obtained with 11 days of admission for COVID-19 (ISARIC). This
acute illness cohort was sub-divided into normal neurologic function (GCS 15) or
neurologic dysfunction (GCS 14 or less). The second cohort consisted of subjects
previously diagnosed with COVID-19 and subsequently diagnosed with onset of a
neurologic disease within 6 weeks of diagnosis (COVID-CNS). These subjects had
serum samples drawn less than 6 weeks from admission (early convalescent) or > 6
254 weeks from admission (late convalescent).

A mouse model to simulate the neuropathology seen in human COVID-19 patients
was also developed by the authors.

These data provide insights into some of the immunological changes that may

mediate neurological dysfunction following COVID-19. However, the data are
tempered by the comments below.

Overall, **this manuscript is difficult to follow at times**. Providing a focused
hypothesis stating the objective of this study would provide readers context for why
this study is important and why specific assays were utilized to address the
hypothesis.

Response: Thank you for asking us to make this more clear to follow- We have
generally formatted the manuscript to explain why the study is important and why
assays were chosen, in addition the hypothesis is now stated in introduction lines 87-
89:

“We tested the hypothesis that immune mediators would correlate with brain injury
markers and reveal a signature of neurological complications associated COVID-19”.

**Methods:**

Two separate patient cohorts impacts the continuity of this study. Furthermore, the
characterization of neurological dysfunction differs between the two cohorts. The
ISARIC cohort neurologic dysfunction was defined as GCS 15 compared to equal to
or less than 14. The COVID-CNS cohort provided greater granularity in defining
specific neurological disorders. Therefore, the continuity of neurological dysfunction
differs from cohort to cohort. The authors should provide a rationale for using two
separate cohorts using different characterizations of neurological dysfunction.

Response: Thank you for highlighting the nature of the two cohorts and we agree
that there is no consistency in their characterizations. Unfortunately, this was a
necessary limitation of many of the very early samples collected in the acute phase
of the pandemic at a time when clinical and research resources were both under
pressure.

The ISARIC study was designed for pandemic preparedness against respiratory
infection. The data collected was centred on respiratory illness. Detailed neurological
complication information was not collected; nevertheless, this is a very valuable
cohort to learn from as they were studied at the very beginning of the pandemic at a
time when research and clinical services were very stretched. This gave us the
opportunity to study acute samples in sick patients. The downside is that we do not
have detailed neurological complication data, as this was not the focus of the ISARIC
study.

We acknowledge the limitations of studying this cohort and were able to establish a
separate cohort where case definitions had been established (Ellul et al. *Lancet
Neurology* 2020)⁷.

The value of the COVID-CNS cohort is that there were clinical case definitions which
had been established, published, and validated. The focus of the COVID-CNS study
was on neurological complication, therefore much more detailed neurological data
was collected. This allowed us to capture the neurological diagnosis, understand the
clinical nature of the diagnosis and to which clinical case definition patients should

be assigned to. We acknowledge that the downside of this is that many of these
samples were necessarily collected during the convalescent phase.

The mouse SARS-CoV-2 infection studies used two different titers for infection (low
vs high inoculation). What data supported the specific concentrations of virus used
for infections? Is there any data correlating these titers to humans with COVID-19
with or without neurological dysfunction?

Why were rat brains, as opposed to mouse, used for incubation with human sera?
What is the data to support cross reactivity of human antibodies from COVID-19
patients with rat brain epitopes?

Line 522 appears to have a place-holder for model details regarding the Leica
confocal microscope.

Response: Very good points and we are very interested in understanding how viral
load affects outcomes. We used the rat brains as this is the conventional screen for
brain-reactive antibodies (detect human IgG bound the brain by IHC) as a first check
of what regions of the brain might be affected by autoantibodies. This method has
been published previously as a way to screen for CNS reactive autoantibodies
(references below).

- Ances, B. M. *et al.* Treatment-responsive limbic encephalitis identified by neu-
322 ropil antibodies: MRI and PET correlates. *Brain* **128**, 1764–1777 (2005).
 - Lai, M. *et al.* Investigation of LGI1 as the antigen in limbic encephalitis previ-
324 ously attributed to potassium channels: a case series. *Lancet Neurol.* **9**, 776–
325 785 (2010).

With regards to the mouse studies, we acknowledge the limitations and all mouse
experiments have been removed from the revised manuscript.

**Results:**

Clinical data from both cohorts is very limited. The focus of this manuscript are
inflammatory changes in COVID-19 patients with neurological dysfunction. However,
there is no epidemiological data regarding premorbid conditions of the subjects
enrolled, nor is there any data regarding what concurrent morbidities that may have
contributed to a subjects alteration in GCS or neurological dysfunction. For example,
how many subjects had hypoxia and/or hypercapnia associated with COVID-19,
which may affect neurologic function. No data was presented regarding confounding
medications that may affect neurological function, such as analgesia, sedation,
antiseizure medications. Very importantly to this study's findings of elevated
inflammatory mediators and autoantibodies, there are no reported data regarding the
administration of immunosuppressive/immunomodulatory medications such as
dexamethasone, remdesivir, or tocilizumab often used for the treatment of
hospitalized COVID-19 patients. The authors briefly mention this in the discussion,
but this really should be added to this manuscript due to potential significant effect

on the regulation and function of immune mediators. It is difficult to accurately
**interpret the results of this study without knowing which subjects received**
**immunosuppressive/immunomodulatory medications.**

Response: The reviewer raises an important point about co-morbidities, treatments
that affect neurological function, and immune modulating therapies. Since the
ISARIC study is a rapid response protocol to monitor respiratory infection,
information on neurological complications is limited. In order to study the
neurological complications, the COVID-Clinical Neuroscience Study recruited COVID
controls and neurological cases with in-depth clinical assessments.

Within the COVID-Clinical Neuroscience Study, the clinical frailty scale scores were
not different between the COVID and Neuro-COVID groups (Mann-Whitney test) and
the co-morbidities did not differ either (Fisher's exact tests). There were very low
numbers of known reports of immunomodulation in both groups as the vast majority
were recruited prior to the introduction of more advanced therapies, such as
remdesivir, tocilizumab etc, in routine practice. Corticosteroids were administered in
56% and 61% for COVID and Neuro-COVID groups, respectively. These are all now
reported in Supplementary Data Table 7 and referenced in line 378 in the methods
section.

In figure 1 early and late convalescent were lumped together in subjects positive for
COVID with or without neurologic disease. Were there differences between early
versus late convalescent subjects?

Response: Thank you for raising this interesting point, we have now made more
clear the difference between early and late convalescent samples. In particular, in
early convalescent samples, NfL and GFAP were elevated in both COVID and
NeuroCOVID vs controls, with a trend towards higher levels in the subset with
NeuroCOVID. Importantly, in the late convalescent samples NfL and GFAP were only
elevated in the NeuroCOVID group, suggesting ongoing neuroglial injury above that
which would be anticipated due to COVID without a neurological complication
(Figure 1m-p).

The autoantibody assays measured both IgM and IgG reactivity from ISARIC sera.
These subjects had their blood drawn within first 11 days of admission. Please
comment on how to tease out IgG autoantibodies associated with the acute COVID
infection from prior illness or antigen exposure. What is the potential effect from prior
environmental antigens?

Response: This is an important point and definitely the IgG could be a result of
previous antigen exposures. We have discussed the hypothesis for this more in lines
293-295.

"The autoantibodies detected in COVID-19, as in other infections, could be through
molecular mimicry or bystander effects³⁶⁻³⁹, but the lack of association of

autoantibody levels with markers of brain injury is evidence against a causal role for
these adaptive immune responses.”

Figure 3 states acute sera containing IgG antibodies against CNS proteins, however
these assays utilized acute samples and measured IgG antibodies instead of IgM
antibodies. The text in the results section states IgG and IgM antibodies were
measure from sera. Please clarify.

Response: We measured both IgM and IgG on the HuProt microarray, we have now
made this clearer by having main figures for both IgM (Figure 3) and IgG (Figure 4)

Mice were infected with low vs high viral titers. Both groups showed viral replication
in the brain, with the lower inoculated mice with less viral replication. Why were only
the low inoculated mice data shown in figure 5? What did the high inoculated mice
show in regards to inflammatory mediators compared to low inoculated mice?

Response: We focused on the low-inoculum infected mice as these did not have
evidence of direct viral infection in the brain—so the effects seen would be from
indirect viral effects.

**The mouse model has been removed from this manuscript.**

**Discussion:**

Regarding lines 291 and 292 in the discussion how do we know that IgG
autoantibodies were associated with SARS-CoV2 infection? This could be due to a
prior environmental antigen since this study is measuring IgG.

Response: The reviewer raises an important point about the timing of assessing the
IgG antibody responses and is correct that the IgG response could just be an
accentuation of previously circulating autoimmune B cells. The fact that the response
is polyclonal also indicates a non-specific inflammatory response. This is now
discussed further in lines 292-295 of the revised manuscript (as above):

“The autoantibodies detected in COVID-19, as in other infections, could be through
molecular mimicry or bystander effects³⁶⁻³⁹, but the lack of association of
autoantibody levels with markers of brain injury is evidence against a causal role for
these adaptive immune responses.”

Line 304 of the discussion states “absence of viral replication in the brain
parenchyma” however SARS-CoV-2 N1 transcript was detected in four of five brains
of mice that had received high inoculum of SARS-CoV-2 and in six of nine that
received low inoculum. Please clarify this statement with data presented in figure 4.

Response: This referred to subgenomic E as a marker of viral replication which was
absent in all the tissue analyzed.

**The mouse model has now been removed from this paper.**

Cytokines were measured from serum samples and not from CSF. What are
potential systemic effects these cytokines have directly on brain constituent cells and
cerebrovasculature? The discussion states the potential effect of injury to the
cerebral vasculature in mediating neurological dysfunction following COVID-19. Why
was brain vascular histology or biomarkers of endothelial glycocalyx/blood-brain
barrier degradation not included since these mice were infected with SARS-CoV-2?
This is especially important in light of data from acutely infected subjects where IgM
may mediate a role in neuronal dysfunction. IgM are pentamers with approximately
molecular weight of 900 kDa, which would require BBB permeability in order to gain
entry into the brain tissue.

Response: It would be very informative to measure cytokines in the CSF. Pro-
inflammatory cytokines can have systemic effects on the cerebrovasculature and
neurons directly. These are important points that remain to be addressed in another
paper. We did check for BBB integrity in the mice and did not find a significant
difference reflecting the mild pathology of this model. Future work in human and
animal models will assess BBB permeability (e.g. by MRI).

**The mouse model has been removed from this manuscript.**

**References**

- 1. Porrett, P. M. Biologic mechanisms and clinical consequences of pregnancy allo-
immunization. *American Journal of Transplantation* **18**, 1059–1067 (2018).
- 2. Ravindranath, M. H. *et al.* Antibodies for β 2-Microglobulin and the Heavy Chains
of HLA-E, HLA-F, and HLA-G Reflect the HLA-Variants on Activated Immune
Cells and Phases of Disease Progression in Rheumatoid Arthritis Patients under
Treatment. *Antibodies* **12**, 26 (2023).
- 3. Rivera-Correa, J. & Rodriguez, A. Autoantibodies during infectious diseases: Les-
sons from malaria applied to COVID-19 and other infections. *Frontiers in Immu-*
*nology* **13**, (2022).
- 4. Mohkhedkar, M., Venigalla, S. S. K. & Janakiraman, V. Untangling COVID-19 and
autoimmunity: Identification of plausible targets suggests multi organ involvement.
*Molecular Immunology* **137**, 105–113 (2021).
- 5. Moody, R., Wilson, K., Flanagan, K. L., Jaworowski, A. & Plebanski, M. Adaptive
Immunity and the Risk of Autoreactivity in COVID-19. *International Journal of Mo-*
*lecular Sciences* **22**, 8965 (2021).
- 6. Johnson, D. & Jiang, W. Infectious diseases, autoantibodies, and autoimmunity.
*Journal of Autoimmunity* **137**, 102962 (2023).
- 7. Ellul, M. A. *et al.* Neurological associations of COVID-19. *The Lancet Neurology*
**19**, 767–783 (2020).
- 8. Needham, E. J. *et al.* Brain injury in COVID-19 is associated with dysregulated in-
nate and adaptive immune responses. *Brain* awac321 (2022)
doi:10.1093/brain/awac321.

- 9. Ross Russell, A. L. *et al.* Spectrum, risk factors and outcomes of neurological and
psychiatric complications of COVID-19: a UK-wide cross-sectional surveillance
study. *Brain Communications* **3**, fcab168 (2021).
- 10. Fernández-Castañeda, A. *et al.* Mild respiratory SARS-CoV-2 infection can
cause multi-lineage cellular dysregulation and myelin loss in the brain.
2022.01.07.475453 Preprint at
<https://www.biorxiv.org/content/10.1101/2022.01.07.475453v1> (2022).

Reviewers' Comments:

Reviewer #2:

Remarks to the Author:

The authors have adequately addressed all the concerns I had raised previously. The data on the mouse model has now been deleted. They now state that the autoantibodies are likely non-specific and not of any pathological significance. At least their data would suggest that since no functional assays have been performed with the antibodies. Please see my previous comments with regards to the strengths of the study.

Reviewer #3:

Remarks to the Author:

The authors have addressed my comments.